# EFFICIENT PERSONALIZATION OF GENERATIVE MODELS VIA OPTIMAL EXPERIMENTAL DESIGN

## ABSTRACT

Preference learning from human feedback has the ability to align generative models with the needs of end-users. Human feedback is costly and time-consuming to obtain, which creates demand for data-efficient query selection methods. This work presents a novel approach that leverages optimal experimental design to ask humans the most informative preference queries, from which we can elucidate the latent reward function modeling user preferences efficiently. We formulate the problem of preference query selection as the one that maximizes the information about the underlying latent preference model. We show that this problem has a convex optimization formulation, and introduce a statistically and computationally efficient algorithm ED-PBRL that is supported by theoretical guarantees and can efficiently construct structured queries such as images or text. We empirically present the proposed framework by personalizing a text-to-image generative model to user-specific styles, showing that it requires less preference queries compared to random query selection.

## 1 INTRODUCTION

**Generative Models & Reinforcement Learning**    In recent years, large-scale generative models have demonstrated tremendous success in generating high-fidelity content across various modalities (Brown et al., 2020; Rombach et al., 2022; Brooks et al., 2024). These models produce content through iterative processes: LLMs generate text token by token (Brown et al., 2020; Ouyang et al., 2022) and diffusion models refine outputs over multiple denoising steps (Dhariwal & Nichol, 2021). For this reason, the Reinforcement Learning (RL) paradigm provides a natural framework for controlling and personalizing these models through feedback mechanisms. Several works have successfully leveraged intermediate feedback during generation: LLMs can be guided through conversational feedback (Ouyang et al., 2022; Christiano et al., 2017), text-to-image models can incorporate human preferences at various generation stages (Lee et al., 2023b; Black et al., 2024), and diffusion models can be steered using reward signals during the denoising process (Fan et al., 2023; Clark et al., 2024). This RL framing enables optimizing policies to produce outputs aligned with learned reward functions, improving generation quality (Lee et al., 2023a; Xu et al., 2023), ensuring safety constraints (Bai et al., 2022; Askell et al., 2021), and personalizing to user preferences (Ouyang et al., 2022; Rafailov et al., 2023; Stiennon et al., 2020).

**Preference-Based RL for Personalization**    Framing the generative process as an RL problem is particularly powerful for personalization, as it allows for aligning the agent's policy with a user's subjective taste. The key challenge is that this taste is difficult to formalize as a numerical reward function. Reinforcement Learning from Human Feedback (RLHF) is the established paradigm for this, learning rewards from human-supplied demonstrations or other forms of feedback (Ziebart et al., 2008; Finn et al., 2016; Linder et al., 2022; Casper et al., 2023). Perhaps the most prominent and practical instance of RLHF is Preference-Based Reinforcement Learning (PBRL), where the latent reward model is learned from comparative feedback (e.g., a user choosing between two generated images). This feedback modality is often more intuitive for humans to provide than absolute scores or full demonstrations (Christiano et al., 2017; Sadigh et al., 2017; Biyik et al., 2019; Ouyang et al., 2022; Saha et al., 2023; Azar et al., 2024). After collecting preference feedback from the user, an estimated reward model then serves as the reward signal aligning the RL agent to the human.

**PBRL Query Selection via OED**    The success of PBRL, however, hinges on the accuracy of this learned reward model, which in turn depends on the quality of the preference queries presented for

Figure 1: The personalization workflow: ED-PBRL calculates policies $\pi_1, \ldots, \pi_K$ which select prompts in the combinatorial token space. These prompts are embedded (CLIP) and rendered with Stable Diffusion 1.4 (Rombach et al., 2022); preferences on the resulting images are collected and used to estimate the guidance model $\hat{\theta}$. Each prompt is formed by a sequence of tokens—a trajectory of a policy $\pi_i$. The $K$ policies are chosen so that, for a given budget, the preferences and embeddings yield an accurate estimate of the guidance model. Notice that each policy can be parameterized via a large table or as a separate generative language model.

user feedback. Collecting these user preferences is a significant practical bottleneck, as it requires a human to provide numerous labels—a process that is both time-consuming and costly (Ouyang et al., 2022; Lee et al., 2023a). This data collection bottleneck makes sample efficiency paramount, which requires selecting maximally informative queries. Existing PBRL methods for selecting such queries often face a trade-off: they are either computationally tractable but lack theoretical guarantees, or they are theoretically grounded but computationally expensive (Chen et al., 2022; Wu et al., 2023; Saha et al., 2023; Zhan et al., 2023; Pacchiano et al., 2023). This raises a fundamental question for making personalization practical:

> *Can we find policies that select queries for preference-based feedback in a way that is both statistically efficient, and computationally tractable?*

In this work, we address this question by leveraging the principles of Optimal Experimental Design (OED) (Chaloner & Verdinelli, 1995; Pukelsheim, 2006; Fedorov & Hackl, 1997). We propose a method to select the most informative queries to present to the user, ensuring that the preference model is learned with as few interactions as possible. Specifically, our objective is to determine a set of $K$ distinct exploration policies for the agent that generate queries. These policies are carefully chosen to generate informative set of queries. When the user provides feedback on these, they provide maximal information about their latent reward parameters. We achieve this by reformulating the generally intractable OED problem (Pukelsheim, 2006; Fedorov & Hackl, 1997) into a continuous optimization problem over the space of state visitation measures induced by the exploration policies. This allows us to use Convex Reinforcement Learning Hazan et al. (2019) to efficiently compute the optimal set of policies for query generation. In tabular settings, we prove that our objective is concave and global optimal solution can be reached.

**Our contributions** We provide the following:

- We formalize the problem of query selection for generative models with Markov processes (Sec. 3), and propose ED-PBRL, a method that builds on Optimal Experimental Design to efficiently solve the problem of learning preferences from a minimal number of queries in the Markov process (Sec. 4).

- A novel upper bound on the MSE matrix of the (regularized) MLE in terms of the (regularized) Fisher information matrix, obtained via a self-concordant analysis (App. B.1).

- Assuming a generative model on discrete space, we provide global convergence guarantees for ED-PBRL based on Convex-RL (Sec. 5.1).

- An experimental evaluation of the proposed method on two types of experiments: synthetic ground truth models and real human-in-the-loop feedback, showcasing promising performance for the personalization of text-to-image models (Sec. 6).

## 2 RELATED WORK

**Generative Model Guidance** Generative models, especially diffusion models Ho et al. (2020); Sohl-Dickstein et al. (2015); Dhariwal & Nichol (2021) and Large Language Models (LLMs), have achieved remarkable success, and often benefit from guidance to align outputs with user preferences. For diffusion models, guidance techniques steer pre-trained models by incorporating preference information, for example, through gradients from an auxiliary classifier (*classifier guidance* Dhariwal & Nichol (2021); Song et al. (2021)) or by leveraging conditional model properties (*classifier-free guidance* Ho & Salimans (2022)). Similarly, LLMs are often guided in a post-training phase to better align with user intent; for instance, InstructGPT Ouyang et al. (2022) uses human feedback to fine-tune models to follow instructions. The effectiveness of these methods often hinges on an accurate underlying preference model. Our work focuses on efficiently learning such preference models to enhance personalized generative model guidance.

**Preference-Based Reinforcement Learning** A key challenge in realizing effective generative model guidance is the accurate and efficient learning of the underlying user preference models. Preference-Based Reinforcement Learning (PBRL) offers a powerful paradigm for this, learning rewards (and thus preference models) from comparative feedback, which is often more intuitive for humans than providing explicit reward values or detailed demonstrations. PBRL's focus on preferences aligns well with capturing nuanced user tastes for guidance. Many PBRL advancements focus on statistical efficiency and regret guarantees Chen et al. (2022); Saha et al. (2023); Zhan et al. (2023); Pacchiano et al. (2023). However, these methods can rely on computationally expensive components, such as oracles for selecting informative queries over pairs of policies from an exponentially large set, or complex algorithmic structures Wu et al. (2023). Our work differs by focusing on a computationally tractable method for query selection in PBRL. We optimize a set of $K$ exploration policies to generate informative comparative queries using an experiment design (ED) objective, rather than relying on pairwise policy comparison oracles. Closest to our goal, Information Directed Reward Learning (IDRL) Lindner et al. (2022) selects queries to disambiguate return differences between a maintained set of plausibly optimal policies; maintaining such a candidate set can be restrictive in large spaces. By contrast, we avoid candidate-policy maintenance entirely by optimizing visitation measures for $K$ exploration policies in a single convex program.

**Optimal Experiment Design** To efficiently learn preference models for guidance, the queries presented to the user must be highly informative. Optimal Experimental Design (OED) Pukelsheim (2006); Fedorov & Hackl (1997) provides principles for selecting experiments to maximize information gain, often by optimizing scalar criteria of the Fisher Information Matrix. Due to the NP-hardness of discrete design, continuous relaxations optimizing over design measures are common. Mutny et al. (2023); Wagenmaker & Jamieson (2022) and Folch et al. (2024) applied OED to active exploration in Markov Chains by optimizing over visitation measures of a single policy. Our work adapts OED to PBRL by designing a *set of K policies* for generating informative *comparative queries*, whilst making it tractable using the framework of convex Reinforcement Learning (Convex-RL) Hazan et al. (2019); Zahavy et al. (2021).

## 3 PROBLEM SETTING

We frame the task of personalized content generation as an RL problem, where the agent sequentially appends to or refines its output. The reward function is unknown and defines the latent personal user's taste. The agent's goal is to learn this latent reward model using the fewest preference queries possible to be given user feedback upon.

### 3.1 MARKOV DECISION PROCESS

To formalize the control of generative models, we employ a finite-horizon Markov Decision Process (MDP) defined by the tuple $\mathcal{M} = (\mathcal{S}, \mathcal{A}, P, H)$. Here, $\mathcal{S}$ and $\mathcal{A}$ represent the state and action spaces, $P(s'|s, a)$ is the *known* transition matrix governing the dynamics of the generation process, and $H$ is the finite horizon. A policy $\pi(a|s)$ defines the agent's strategy for making sequential choices.

**Examples of MDPs for Generative Models** Our abstract MDP formulation is intentionally general. To make this concrete, we provide several examples of how it can be instantiated:

*Automatic Prompt Engineering*: The sequential construction of prompts for text-to-image models can be modeled as an MDP, where states are steps in the token sequence and actions are

choices of next tokens to append to the prompt. This is the formulation that we use in our experiments in Section 6.

*Autoregressive Text Generation*: For an LLM, a state is the sequence of tokens generated so far, and an action is selecting the next token from the vocabulary.

*Iterative Refinement in Diffusion Models*: A state can be the noisy data (e.g., an image) at a denoising step, and an action could be selecting a guidance direction for refinement.

## 3.2 LATENT REWARD AND PREFERENCE FEEDBACK

We model the user's latent reward as linear in known features: a map $\phi : \mathcal{S} \times \mathcal{A} \to \mathbb{R}^d$ embeds state-action pairs, and $r(s, a) = (\theta^*)^\top \phi(s, a)$ for an unknown $\theta^* \in \mathbb{R}^d$. The goal is to estimate $\theta^*$ with $\hat{\theta}$ efficiently from preferences. The framework and guarantees extend to richer classes (e.g., RKHS linear functionals, c.f. Mutny et al. (2023)). For clarity the reward is only state dependent, e.g. $r(s) = (\theta^*)^\top \phi(s)$. This is without loss of generality as state-action rewards are recovered by augmenting state space as $\mathcal{S}' = \mathcal{S} \times \mathcal{A}$.

**Preference model** To learn $\theta^*$, we rely on comparative feedback. Given $K$ options (e.g. states, actions or trajectories), denoted by $\{x_1, \ldots, x_K\}$, a user selects the one they prefer most. We model the probability of this choice using the standard multinomial logit (softmax) model. The probability that a user chooses option $x_q$ is then:

$$\mathrm{P}(x_q \text{ is best}) = \frac{\exp((\theta^*)^\top \phi(x_q))}{\sum_{k'=1}^K \exp((\theta^*)^\top \phi(x_{k'}))} \tag{1}$$

where $\phi(x'_k)$ is the feature vector of option $x'_k$. This model is a generalization of the Bradley-Terry model (which corresponds to $K = 2$) (Bradley & Terry, 1952).

## 3.3 INTERACTION PROTOCOL

The learning process follows a fixed experimental design protocol with three phases:

1. **Policy Optimization:** The algorithm determines a set of $K$ exploration policies, $\pi_1, \ldots, \pi_K$, by solving an information-maximization optimization problem (detailed in Section 4).

2. **Data Collection:** The $K$ policies are executed for $T$ episodes, generating $T$ sets of trajectories. Each set is $\{\tau_{t,1}, \ldots, \tau_{t,K}\}$, where $\tau_{t,q} \sim \pi_q$. These sets (or their components, see below) are presented to the user, who provides one preference choice at each of the $H$ timesteps, resulting in a dataset of $T \times H$ feedback comparisons.

3. **Parameter Estimation:** Using the collected feedback and the features of the corresponding trajectories, the algorithm computes the final estimate $\hat{\theta}$ of the true parameter $\theta^*$.

The central challenge, which we address, is how to perform Phase 1 to select policies that make the estimation in Phase 3 as efficient as possible. The pipeline is visualized in Figure 1 with image feedback generated via prompts (trajectories).

## 3.4 FEEDBACK MODELS

We consider two plausible models for how feedback is elicited over the generated trajectories.

**State-based Preference Feedback** At each timestep $h \in [H]$, the user compares states $\{s_{1,h}, \ldots, s_{K,h}\}$ from the $K$ trajectories. The choice probability is given by Eq. 1 using state features $\phi(s_{q,h})$. This model is mainly used for our theoretical analysis.

**Truncated Trajectory Preference Feedback** More practically, the user compares partial outputs. The options $x_q$ are trajectories truncated at step $h$, which we denote as a sequence of states $\tau_q[1 : h] = \{s_{q,1}, \ldots, s_{q,h}\}$. For example, in prompt generation, users compare partial prompts like *"A painting of..."* vs. *"A photo of..."*. The choice probability is again given by Eq. 1, but using features of the partial sequence, $\phi(\tau_q[1 : h])$. These features (e.g., a CLIP embedding of a partial sentence) are not necessarily simple sums of their constituent state features. We use this model in our experiments.

### 3.5 ESTIMATION

Given a dataset of $T \times H$ preferences, the parameter $\theta^*$ is estimated via regularized maximum likelihood. Let $y_{t,h,q}$ be a one-hot indicator that alternative $q$ was chosen at step $h$ of episode $t$. The probability of this choice, $p(q|t, h, \theta)$, is given by the softmax model from Eq. 1 applied to the features of the options presented under the relevant feedback model. The estimate $\hat{\theta}$ is the solution to

$$\hat{\theta} = \arg\max_{\theta \in \mathbb{R}^d} \sum_{t=1,h=1,q=1}^{T,H,K} y_{t,h,q} \log \left( p(q_{t,h} \mid t, h, \theta) \right) - \frac{\lambda}{2} \|\theta\|_2^2. \tag{2}$$

where $\lambda \geq 0$ is a regularization coefficient.

## 4 OPTIMAL EXPERIMENTAL DESIGN FOR PREFERENCE LEARNING

Our main motivation is selecting queries for PBRL in a sample efficient manner. *Given a budget $T$, how should we select $K$ exploration policies to generate $T$ sets of $K$ trajectories that are maximally informative for estimating $\theta^*$?* To address this, we use an information-theoretic approach, leveraging the Fisher Information Matrix.

### 4.1 FISHER INFORMATION AND ESTIMATION ERROR

The quality of the estimate $\hat{\theta}$ is fundamentally linked to which queries are selected. The Fisher Information Matrix (FIM), $I(\theta)$, quantifies the information content of the data; classically, the Cramér–Rao Lower Bound (CRLB) relates $I(\theta)$ to a *lower* bound on the covariance of unbiased estimators. In our setting, we establish a *novel upper bound* for the regularized MLE—derived via a self-concordant analysis of the (regularized) log-likelihood: the Mean Squared Error (MSE) matrix of $\theta_\lambda$ is controlled by the inverse of the regularized FIM at the true parameter, $I_\lambda(\theta^*)^{-1}$. This is formalized in the following result.

**Theorem 4.1** (Maximizing FIM improves Estimation). *Under mild conditions (Appendix B.1), the expected square error of $\hat{\theta}_\lambda$, of multinomial likelihood, satisfies*

$$\mathbb{E}[(\hat{\theta}_\lambda - \theta^*)(\hat{\theta}_\lambda - \theta^*)^T] \preceq C_{\theta^*}^\lambda \cdot I_\lambda(\theta^*)^{-1}$$

*where $C_{\theta^*}^\lambda = (1 - r_{\theta^*}^\lambda)^{-4}$ depends on a local consistency radius $r_{\theta^*}^\lambda \in [0, 1)$.*

Thus, maximizing $I_\lambda(\theta^*)$ (making its inverse smaller in Loewner order) reduces estimation error; see Appendix B.1 for the full proof.

### 4.2 THE INTRACTABLE IDEAL OBJECTIVE

Theorem 4.1 motivates maximizing the FIM $I_\lambda(\theta^*)$ to minimize estimation error. Our goal is therefore to select policies that generate trajectories yielding the most information. However, optimizing the FIM over a discrete set of $K \times T$ trajectories is typically NP-hard. A standard OED approach is to relax this problem by optimizing over a *distribution* of experiments—in our case, policies that generate trajectories. This leads to optimizing the *expected* FIM that the policies induce Pukelsheim (2006); Fedorov & Hackl (1997).

The expected regularized FIM for $K$ policies $\pi_{1:K}$ generating $T$ episodes is:

$$I_\lambda(\pi_{1:K}, \theta) = T \sum_{h=1}^{H} I_h(\pi_{1:K}, \theta) + \lambda I_d, \tag{3}$$

where $I_h(\pi_{1:K}, \theta)$ is the FIM contribution from timestep $h$, averaged over the trajectory distributions $\eta_{\pi_q}$. Let $s_h^q$ be the state of trajectory $\tau_q \sim \eta_{\pi_q}$ at step $h$. The expression for $I_h$ is an expectation over the FIM for a single multinomial choice (derived in Appendix B.2):

$$I_h(\pi_{1:K}, \theta) = \mathop{\mathbb{E}}_{\substack{\tau_q \sim \eta_{\pi_q} \\ q \in [K]}} \left[ \sum_{q=1}^{K} p(q|h) \, \phi(s_h^q)\phi(s_h^q)^\top - \Big( \sum_{q=1}^{K} p(q|h) \, \phi(s_h^q) \Big) \Big( \sum_{l=1}^{K} p(l|h) \, \phi(s_h^l) \Big)^\top \right]. \tag{4}$$

where $p(q|h)$ is the softmax choice probability for the states $\{s_h^1, \ldots, s_h^K\}$, as defined in (1). The ideal policy then optimizes the objective: $\arg\max_{\pi_{1:K}} s\left( I_\lambda(\pi_{1:K}, \theta) \right)$. However, this ideal objective presents two major challenges:

- **Dependence on unknown** $\theta$: The FIM depends on the true $\theta$, which is unknown at the design stage.
- **Intractable Optimization**: The objective involves an expectation over an exponentially large trajectory space ($\mathcal{S} \times \mathcal{S} \cdots \times \mathcal{S}$, $H$ times) of $K$ independent policies.

### 4.3 REFORMULATION TO A TRACTABLE OBJECTIVE

We address these issues by deriving a tractable objective in three steps (full details in App.B.4).

**Step 1: Reformulation using State Visitation Measures.** We rewrite the expected FIM in terms of state visitation measures (occupancy measures), turning policy expectations into expectations over $s \sim d_{\pi_q}^h$. This yields a design objective over visitation measures; see Lemma B.2 and Appendix Sec. B.4 for the explicit form and the Step 1 problem statement.

**Step 2: $\theta$-agnostic approximation.** The per-step information depends on the unknown $\theta$ through the choice probabilities $p(\cdot)$ (see Appendix Sec. B.4 for the explicit expression). To decouple design from $\theta$ without assuming the worst case $\theta$, we adopt an average-case approach and replace $p(q \mid h; s_{1:K}, \theta)$ by its expectation under an uninformative prior, giving $p(q \mid h; s_{1:K}) \approx 1/K$. For $K = 2$ this holds exactly with a Gaussian prior on $\theta$; for $K > 2$ it is a reasonable approximation when alternatives are diverse/symmetric. Substituting yields a $\theta$-independent surrogate $\tilde{I}_h(d_{1:K}^h)$ (Appendix Eq. 10).

**Step 3: Marginalization** The expectation in the approximate FIM (Eq. 10) with the state-visitation be resolved into a tractable matrix form, decomposable to per-timestep contribution $\tilde{I}_h(d_{1:K}^h)$ as:

$$\tilde{I}_h(d_{1:K}^h) = \Phi^T \left( \frac{1}{K} \sum_{q=1}^{K} \mathrm{diag}(d_q^h) - \bar{d}^h(\bar{d}^h)^T \right) \Phi, \tag{5}$$

where $\Phi$ is the matrix of concatenate state features, $d_q^h$ is the visitation vector for policy $q$ at step $h$, and is the average visitation $\bar{d}^h = \frac{1}{K} \sum_q d_q^h$. We show this formally in Theorem B.3. The resulting approximated information matrix in Eq. (5) is a function of $|S| \times H \times K$ variables for discrete state spaces and can be efficiently optimized upon proper scalarization.

**Scalarization** The objective (5) is the matrix-valued function of $d_{1:K}$. As such, it cannot be optimized, since the information about different components of $\theta_i$ of the latent reward needs to be weighted. Classically, experiment design Fedorov & Hackl (1997) suggest to either weight all components of the uncertainty equally such as reducing $||\hat{\theta} - \theta^\star||^2$, or minimizing error on a certain projection $(\hat{\theta} - \theta^\star)^\top \mathbf{V}(\hat{\theta} - \theta^\star)$. These are called A- or V-designs, respectively, and result in a particular scalarization function $s(\cdot)$. With scalarization our final design objective is then:

$$\underset{d_{1:K} \in \mathcal{D}}{\arg\max} \ s(I_{\mathrm{total}}(d_{1:K})) \ := \ \underset{d_{1:K} \in \mathcal{D}}{\arg\max} \ s\left( T \sum_{h=1}^{H} \tilde{I}_h(d_{1:K}^h) + \lambda I_d \right). \tag{6}$$

For A-design this is $s(\mathbf{A}) = 1/\mathrm{Tr}(\mathbf{A}^{-1})$, and for V-design, $s(\mathbf{A}) = 1/\mathrm{Tr}(\mathbf{V}\mathbf{A}^{-1})$. This optimization is subject to the constraints that each $d_q^h$ must be a valid visitation of policy $\pi_q$ (i.e. $\in \mathcal{D}$).

### 4.4 INFORMATION RELATIONSHIP OF FEEDBACK MODELS

The objective (Eq. 6) is derived for the state-based feedback model. Arguably, the more practical variant is the truncated trajectory feedback model. In the following result, we provide a link between state-based and truncated-trajectory-based feedback using additive feature decomposition assumption, and prove that by optimizing a policy for state-based feedback design in Eq. 6 we are also improving the truncated-trajectory design with the same policy.

**Theorem 4.2** (Truncated trajectory). *If the features of the partial trajectory admit additive decomposition, i.e. $\phi(\tau[1:h]) = \sum_j^h \phi(s_j)$, the approximate Fisher Information (FI) of truncated trajectory feedback is lower-bounded by the FI of the state-based feedback scaled by $1/4$.*

This result is formally stated and proven as Theorem B.5 in Appendix B.5.

## 5 THE ED-PBRL ALGORITHM AND GUARANTEES

In this section, we summarize the optimization and the interaction protocol. In particular, we present a *static* version of ED-PBRL algorithm. We first determine $K$ policies by maximizing the objective

in Eq. 6. Here $d_{1:K} = \{d_q^h\}_{q \in [K], h \in [H]}$ are (state) visitation measures of the K policies. These then generate trajectories (e.g. prompts) for preference collection, after which we fit $\hat{\theta}$. See Alg. 1.

---

**Algorithm 1** ED-PBRL (Conceptual Overview)

---

**Input:** MDP details $(M, \Phi)$, design parameters $(K, H, T, s(\cdot), \lambda)$
**Output:** Estimated preference parameter $\hat{\theta}$
**Phase 1: Compute Optimal State Visitation Measures**
    Solve Eq. 6 for optimal visitation measures $\{d_q^{*h}\}_{h,q}$.
**Phase 2: Policy Extraction and Trajectory Sampling**
    Extract policies $\{\pi_q^*\}_{q=1}^K$ from $\{d_q^{*h}\}_{h,q}$.
    Sample $K \times T$ trajectories using $\{\pi_q^*\}$ and collect preference feedback.
**Phase 3: Parameter Estimation**
    Estimate $\hat{\theta}$ using all collected feedback (cf. Section 3).

---

The solver in *Phase 1* uses the Frank–Wolfe algorithm, which sequentially linearizes the objective and solve the linearized problem over the visitation measures. Let $x \in \mathcal{S}$ denote a state, and let $d_{1:K}$ collect the visitation measures for all policies. The algorithm update the policies in iterative fashion, at iterate $n$, for each policy $q$ define the per-policy linearization of Eq.(6) $G_q^{(n)}(x) :=$ $\nabla_{d_q(x)} s(I_{\text{total}}(d_{1:K}^{(n)}))$, evaluated at the current iterate $d_{1:K}^{(n)}$. The linearization decouples across policies and can be solved separately. For a single policy $q \in [K]$, denote by $d_{\pi_q}$ its visitation vector. The linearization oracle for this policy solves:

$$\Delta d_{\pi_q}^{n+1} = \arg\max_{d \in \mathcal{D}} \sum_{\mathcal{S}} d(x) G_q^{(n)}(x)$$

where $\mathcal{D}$ is the convex set of valid visitation measures, and $\sum$ turns to $\int$ for continuous state spaces. The linear oracle problem exactly coincides with the reinforcement learning problem where $G_q^{(n)}(x)$ plays the role of reward, a key observation from the Convex-RL (Hazan et al., 2019; Zahavy et al., 2021). The next iterate is constructed using convex combination $d_{\pi_q}^{(n+1)} = \alpha_{n+1} d_{\pi_q}^{(n)} + (1 - \alpha_{n+1}) \Delta d_{\pi_q}^{n+1}$ chosen via line search oracle. After sufficient number of steps, we extract policies $\pi$ via marginalization $\pi_q^*(a \mid s) \propto d_{\pi_q}^*(s, a)$. Afterwards in *Phase 2* and *3*, we sample $K \times T$ trajectories and estimate $\hat{\theta}$. See Appendix Alg. 2 for more detailed description.

### 5.1 THEORETICAL GUARANTEES AND ALGORITHM VARIANTS

Frank–Wolfe (FW) is known to provably converge to maximum on concave objectives defined on convex sets for well chosen sequence of $\alpha_n$. The set of visitation measures is convex as it is a polytope, and we show the objective is concave.

**Theorem 5.1.** *Let $d_{1:K} = \{d_q^h\}_{h \in [H], q \in [K]}$. If the scalarization $s : \mathbb{S}_+^d \to \mathbb{R}$ is concave and matrix-monotone, then $(I_{total}(d_{1:K}))$ as defined in (6) is concave in $d_{1:K}$.*

FW converges to the global optimum with the standard $\mathcal{O}(1/n)$ rate under an exact linear oracle (Jaggi, 2013). Detailed statements and proofs (including the oracle instantiation and concavity proof) are provided in Appendix Sec. B.7.

**Adaptive variant**  The *static* version does not use partial feedback from prior queries to update policies $\pi_q$ $q \in [K]$. We can run, an *adaptive* variant where collect a batch of preferences, update $\hat{\theta}$, then re-optimize visitation and hence policies $\pi_q$. Round 1 uses $p \approx 1/K$, but with adaptive variant we could use an estimated $\hat{p}(q \mid h)$ (e.g. via $\hat{\theta}$) inside the objective and refine intermediate visitation measures.

## 6 EXPERIMENTAL EVALUATION

We evaluate our framework to personalize text-to-image generation based on CLIP embeddings Radford et al. (2021). We conduct two types of experiments: (1) a quantitative evaluation using synthetic ground truth (GT) models to simulate user preferences, and (2) a human study with multiple participants. The experimental workflow is shown in Figure 1.

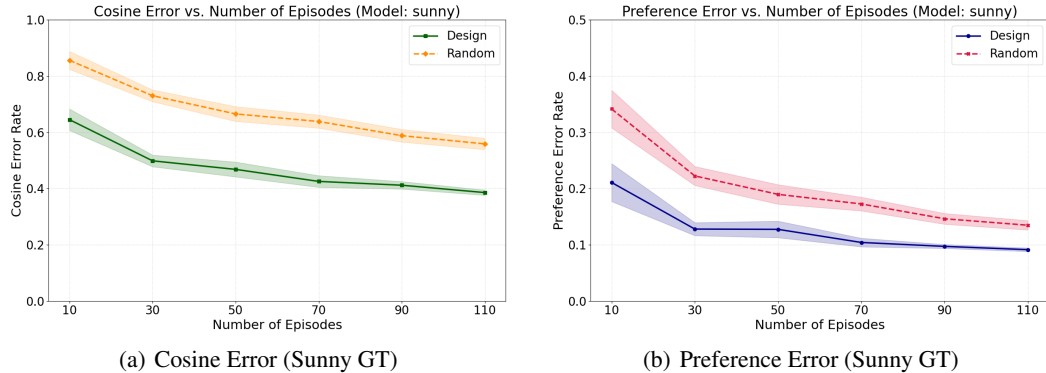

(a) Cosine Error (Sunny GT)  (b) Preference Error (Sunny GT)

Figure 2: Performance of ED-PBRL on the Sunny synthetic Ground Truth (GT) model. We plot the Cosine Error (left) and Preference Prediction Error (right) against the number of interaction episodes. These results demonstrate the efficiency of our OED approach. Numerical results for all GT models (Sunny, Medieval, and Technological) are presented in Appendix (Figure 5).

**Experimental Methodology** We briefly comment on the overall setup; full details are in Appendix Sec. A. We construct a prompt in finite-horizon MDP; states are timesteps ($H = 6$), actions are design tokens from a fixed vocabulary, and a trajectory yields a prompt. We randomly select an initial *base-prompt*, which represent the overall content of the image. Token and prompt features are embedded using CLIP, and preferences are modeled linearly via the Truncated Trajectory Feedback model (see Sec. 3.4). Design is optimized with the V-design scalarization (Sec. 4.3) where the matrix $V$ contains concatenated prompt embeddings of base prompts (see App. A.1 for details). A consolidated list of hyperparameters appears in Appendix Table 1.

## 6.1 Synthetic Ground Truth Model Experiments

We perform quantitative evaluation of ED-PBRL against known ground truth (GT) preference models. We simulate a user whose preferences are dictated by a GT linear preference model $\theta^*$. Each GT model is constructed from the normalized CLIP text embedding of a descriptive sentence. For instance, the *Sunny GT* model, which is the focus of our main results, uses the phrase *"An image with warm colors depicting bright sunshine"*. We also evaluate against *Medieval and Technological GT* model, with full details for all models provided in Appendix A.2. The goal is to measure how accurately and efficiently our method recovers $\theta^*$. We report two metrics (Appendix A.2): **Cosine Error** — the cosine distance between the learned preference vector $\hat{\theta}$ and the GT vector $\theta^*$; and **Preference Prediction Error** — the error rate of $\hat{\theta}$ in predicting the synthetic user's preference on unseen pairs of prompts from a held-out test token set.

Figure 2 presents the learning curves for these metrics for the *Sunny GT* model, averaged over multiple independent runs. Similar trends hold across the remaining models (see Appendix Figure 5).

## 6.2 LLM-Simulated Preference Experiment

To systematically evaluate our approach across many experimental conditions, we use an LLM (GPT-4.1-mini) as a simulated preference oracle. The LLM receives the same questionnaire-style queries that would be shown to human participants: four candidate images and a style description, with the task of selecting the image that best matches the specified style. This setup enables comprehensive evaluation across multiple regularization strengths, training set sizes, and aesthetic styles—a scale that would be prohibitively expensive with human participants.

**Setup** We evaluate ten target style conditions spanning diverse aesthetics: futuristic, sunny, forest, landscape, ancient, noir, watercolor, cyberpunk, minimalist, and medieval. For each style, the LLM receives a descriptive prompt (e.g., "Futuristic style with advanced technologies") and selects among $K = 4$ candidate images at each step $h \in \{1, \ldots, H\}$ (with $H = 6$). We systematically vary regularization $\lambda \in \{0.1, 1, 10, 100\}$ and training episodes $\in \{10, 20, 30, 40, 50\}$, with 10 held-out test episodes. Each configuration is repeated across 3 cross-validation folds.

| Base Image | ED-PBRL based | Random Exp. based |
|---|---|---|

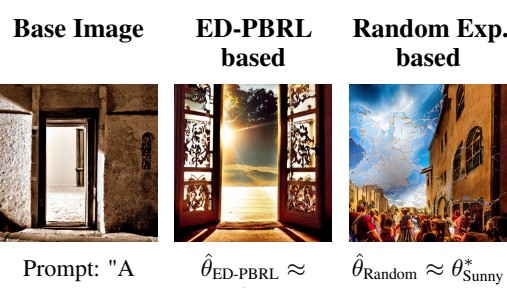

The results show that queries selected ED-PBRL consistently leads to better estimation of the underlying preference model than random exploration, as evidenced by lower error rates. Figure 3 provides a qualitative understanding of these results, illustrating image generation guided by a model estimated by ED-PBRL versus a model learned from random exploration. The ED-PBRL-guided image better reflects the target "sunny" aesthetic.

Prompt: "A photo of a gate"    $\hat{\theta}_{\text{ED-PBRL}} \approx \theta^*_{\text{Sunny}}$    $\hat{\theta}_{\text{Random}} \approx \theta^*_{\text{Sunny}}$

Figure 3: Sunny GT qualitative personalization and experimental context.

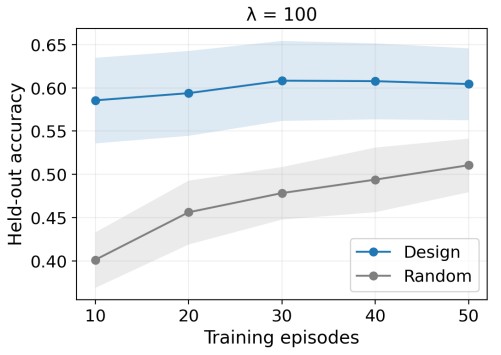
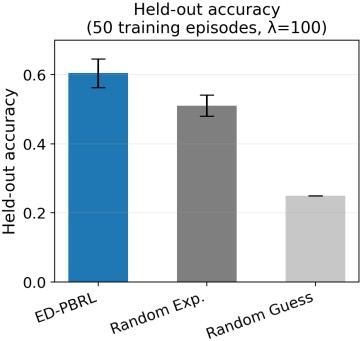

(a) Held-out accuracy vs. training episodes ($\lambda = 100$)     (b) Summary at 50 training episodes

Figure 4: LLM-simulated preference held-out accuracy. Panel (a) shows accuracy as we vary the number of training episodes at $\lambda = 100$, with shaded standard errors across 10 styles. ED-PBRL maintains $\sim 60\%$ accuracy across all training sizes, while random exploration degrades significantly with less data. Panel (b) summarizes results at 50 training episodes. The random-guess baseline is $1/K = 25\%$.

**Evaluation Metric**   We evaluate models by how well the learned $\hat{\theta}$ predicts held-out LLM choices. The held-out benchmark consists of 10 episodes ($10 \times H = 60$ decisions). We report the held-out preference accuracy: the fraction of test decisions where $\arg\max_q \hat{\theta}^\top \phi(\tau_q[1 : h])$ matches the LLM's choice. Results are averaged across styles and folds. Figure 4(a) shows accuracy versus training episodes at $\lambda = 100$, while Figure 4(b) summarizes the accuracy at 50 training episodes.

## 7 CONCLUSION

We introduced ED-PBRL, a novel framework for efficiently personalizing generative models by learning user preference model from a minimal number of *comparative multinomial queries*. Our work demonstrates that the principles of Optimal Experimental Design (OED) can be practically and effectively applied to Preference-Based Reinforcement Learning (PBRL) for modern applications. Namely, we established a practical connection between OED and PBRL for personalizing generative models by framing query selection as an information-maximization problem. ED-PBRL significantly accelerates the learning of a user's latent reward function. This method opens avenue to personalize genAI tools with few interactions compared to standard random query selection and automates the curation of such questions through an expert.

IMPACT STATEMENT

This paper studies ML algorithm that influences data collection. Such algorithms may generate long-term biases in collected datasets and need to be used cautiously. Since these algorithms construct queries with which real users interact, there needs to be a comprehensive framework including ethics standards to ensure they are not causing harm to humans. There are many potential consequences of our work if we use it in products to personalize generative models. Our study is only a proof-of-concept visualizing formalism to perform such personalization. The aspect of bias needs to be further investigated before any deployment, which was out of the scope of this work.

ETHICS STATEMENT

This paper uses LLM-simulated preferences (GPT-4.1-mini) rather than human participants for the preference learning experiments. This approach avoids ethical concerns related to human subject research while enabling systematic evaluation across many experimental conditions. The LLM receives the same questionnaire-style queries that would be shown to human participants, making the evaluation protocol directly transferable to human studies in future work.

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

## A  APPENDIX: DETAILED EXPERIMENTAL SETUP

This section provides a comprehensive description of the experimental environment, parameters, and models used in our evaluation, intended for reproducibility and completeness. The overall workflow is depicted in Figure 1. A summary of key parameters is available in Table 1.

### A.1  COMMON EXPERIMENTAL COMPONENTS

**Environment: Prompt Construction MDP**   The environment is modeled as a finite-horizon Markov Decision Process (MDP) designed to simulate the construction of textual prompts.

- **States ($\mathcal{S}$):** States $s \in \{0, 1, \ldots, H - 1\}$ directly correspond to the current timestep or depth in the prompt construction process.

- **Horizon ($H$):** The horizon corresponds to the number of vocabulary files used for sequential token selection.

- **Actions ($\mathcal{A}$):** Actions are indices corresponding to unique "design tokens" extracted from the vocabulary files. These tokens represent semantic concepts (e.g., "Man sitting", "artistic", "happy").

- **Vocabulary:** The vocabulary is sourced from $H$ files: 'bases.txt', 'ambient.txt', 'style.txt', 'composition.txt', 'lighting.txt', 'detail.txt'. The selection of tokens is structured by timestep. At $s = 0$, only "base" concepts are allowed. For $s > 0$, tokens from other categories are used.

- **Transitions ($P$):** Deterministic. Selecting a token at state $s$ transitions to state $s + 1$.

- **Feature Representation for OED ($\phi(a)$):** The features for design tokens are their 768-dimensional, normalized CLIP text embeddings ('ViT-L/14').

Table 1: Summary of experimental parameters.

| Common Parameters | | | |
|---|---|---|---|
| Feedback Model | Truncated Trajectory | OED Criterion | V-design (App. A.1) |
| Horizon ($H$) | 6 | Frank-Wolfe Iters ($N$) | 100 |
| Num. Policies ($K$) | 4 | FW Step Size | Line Search |
| CLIP Model | ViT-L/14 | | |

| Synthetic Experiment | | LLM-Simulated Preference Experiment | |
|---|---|---|---|
| Feedback Source | GT Model | Feedback Source | GPT-4.1-mini |
| Num. Episodes ($T$) | 10, 30, ..., 110 | Num. Episodes ($T$) | 60 |
| | | | Training $\{10, 20, 30, 40, 50\}$ / |
| Num. Runs | 25 | Episode Split | Benchmark 10 |
| Num. Test Prompts | 1000 | Cross-Validation | 3 folds |
| Num. Eval Pairs | 5000 | Regularization ($\lambda$) | 0.1, 1, 10, 100 |
| Vocabulary Split | 75% train / 25% test | Num. Styles | 10 |
| Regularization ($\lambda$) | 100 | Evaluation Metrics | App. A.3 |
| Evaluation Metrics | App. A.2 | | |

**Preference Model and Estimation**

- **Feedback Model:** We use the Truncated Trajectory Feedback model (Section 3.4) for both experiments. At each timestep $h$, a preference is given over $K$ partial prompts $\{\tau_1[0 : h], \ldots, \tau_K[0 : h]\}$.

- **Features for Estimation ($\phi$(partial prompt)):** The feature vector for a partial prompt is its normalized CLIP text embedding ('ViT-L/14').

**Experimental Design (OED)**    The experimental design objective is to select policies that maximize information about $\theta$.

- **Scalarization Criterion** $s(\cdot)$: We use an A-optimality variant, $s(I_{total,reg}) = -\text{Tr}\left(V(I_{total,reg})^{-1}\right)$, where $I_{total,reg}$ is the regularized total approximate FIM from Eq. 6.

- $V$ **Matrix Construction:** The matrix $V = C^T C$ is constructed from differences between feature embeddings of tokens from the same thematic category (excluding 'bases.txt'), i.e., $c_k^T = (\phi(a_i) - \phi(a_j))^T$. This V-design criterion directly targets the precision of estimated preference differences, which is essential for learning an effective ranking model. The full construction is detailed in the original appendix text.

- **Optimization:** The Convex-RL procedure (Algorithm 2) is used to solve the design problem.

- **Computational Cost:** The one-time design optimization for a vocabulary of approximately 5000 tokens takes around 10 minutes on a single NVIDIA A100 GPU.

## A.2    SYNTHETIC GROUND TRUTH MODEL EXPERIMENTS: SETUP AND METRICS

**Ground Truth Scorer Models**    For the synthetic experiments, we simulate user preferences using three distinct ground truth (GT) scorer models. Each is represented by a weight vector $\theta^* \in \mathbb{R}^d$ constructed by taking the normalized CLIP text embedding of a descriptive sentence:

- **Sunny GT Model** ($\theta^*_{sunny}$): From `CLIP("An image with warm colors depicting bright sunshine")`.

- **Medieval GT Model** ($\theta^*_{medieval}$): From `CLIP("An image with ancient kingdom depicting medieval times")`.

- **Technological GT Model** ($\theta^*_{technological}$): From `CLIP("An image with advanced technologies depicting futuristic style")`.

The GT vector $\theta^*$ is used to simulate user choices and serves as the ground truth for evaluation.

**Evaluation Metrics**

- **Cosine Error:** $1 - \text{cosine\_similarity}(\hat{\theta}, \theta^*)$. Measures the angular deviation between the estimated preference vector and the ground truth $\theta^*$.

- **Preference Prediction Error:** The fraction of pairs where $\hat{\theta}$'s prediction mismatches the GT's preference on prompts generated exclusively from the held-out testing vocabulary.

## A.3 LLM-SIMULATED PREFERENCE EXPERIMENT: SETUP AND METRICS

**Setup** To enable systematic evaluation across many experimental conditions, we use GPT-4.1-mini as a simulated preference oracle. The LLM receives the identical questionnaire-style queries that would be shown to human participants: at each decision point, it sees four candidate images and a style description, and must select the image that best matches the specified style. This approach enables comprehensive evaluation across multiple regularization strengths ($\lambda \in \{0.1, 1, 10, 100\}$), training set sizes ($\{10, 20, 30, 40, 50\}$ episodes), and ten aesthetic styles—a scale that would be prohibitively expensive with human participants.

**LLM Query Format** The LLM receives the following prompt structure for each preference query:

```
You are an art reviewer.
Follow the style guidance below
and choose the single best preference label.
Return only a strict JSON object
in the format {"preference": "<label>"}.
Do not include explanations or additional fields.

Style guidance:
[STYLE DESCRIPTION]
```

The style description (e.g., "Futuristic style with advanced technologies") is substituted for each of the ten target styles. The four candidate images are presented, and the LLM returns its preference.

**Cross-Validation Protocol** For robust evaluation, we use 3-fold cross-validation with rotating test windows. With 60 total episodes per style-algorithm pair, each fold uses 50 episodes for training and 10 for testing. The test window rotates across folds: fold 0 tests on episodes 50–59, fold 1 on episodes 40–49, and fold 2 on episodes 30–39.

**Evaluation Metric**

- **Hold-out Preference Accuracy:** We measure how well the learned model predicts the LLM's choices on unseen data. This is the percentage of times that the preference predicted by $\hat{\theta}$ (i.e., $\arg\max_q \hat{\theta}^\top \phi(\tau_q[1:h])$) matches the actual choice made by the LLM on the 10 held-out test episodes. With a horizon of $H = 6$, this evaluation is performed over $10 \times 6 = 60$ preference decisions per fold, aggregated across 3 folds and 10 styles.

## A.4 FULL NUMERICAL AND QUALITATIVE RESULTS

This section provides the full set of results for all experiments.

**Synthetic Experiment Results**

**Qualitative Results Summary (Synthetic)** For each Ground Truth (GT) model (Sunny, Medieval, Technological), Figures 6, 7, and 8 show a visual comparison of the prompts generated by ED-PBRL (Design) and Random exploration. The figures correspond to the median cosine error run (out of 25 seeds) after $T = 110$ feedback episodes with $K = 4$ policies. To test generalization, the personalized prompts are constructed by adding style tokens selected from the held-out test vocabulary to a base prompt.

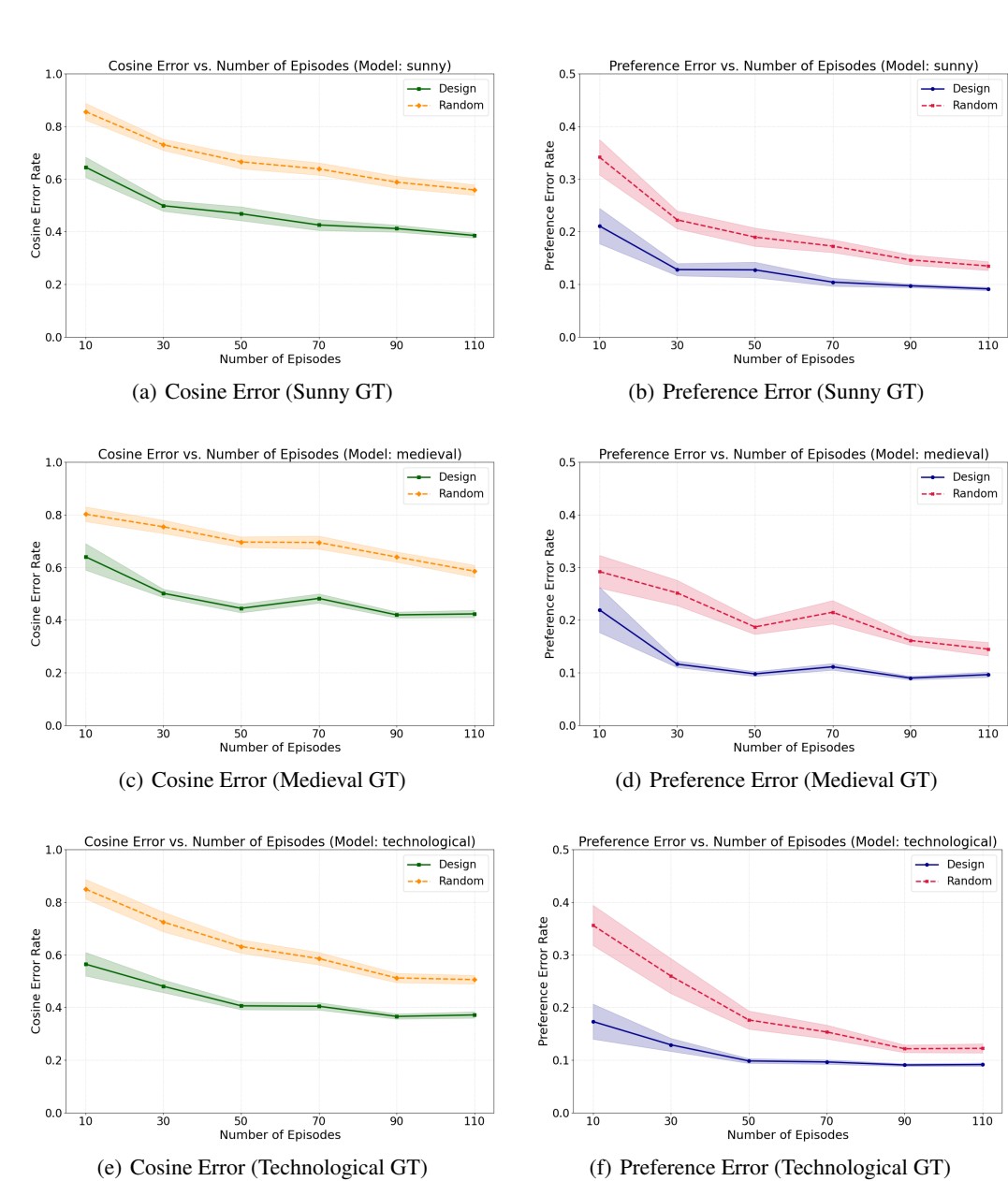

Figure 5: Performance of ED-PBRL on Sunny, Medieval, and Technological synthetic Ground Truth (GT) models. For each GT model, we plot the Cosine Error (left column) and Preference Prediction Error (right column) against the number of interaction episodes. Results are averaged over N=25 independent runs, and the shaded regions represent the standard error of the mean. The Sunny GT model results are also shown in the main paper (Figure 2).

**Top Generated Prompts**

| Base Prompt | Best 1 RankScr: 0.17 GTScr: 0.67 | Best 2 RankScr: 0.17 GTScr: 0.68 | Best 3 RankScr: 0.17 GTScr: 0.69 | Best 4 RankScr: 0.17 GTScr: 0.67 |

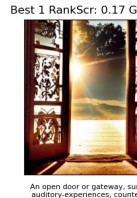 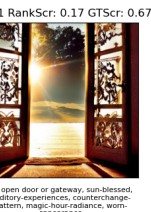 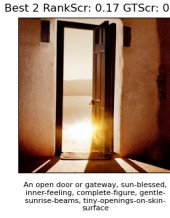 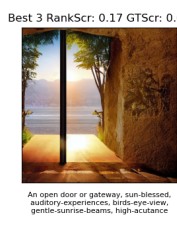 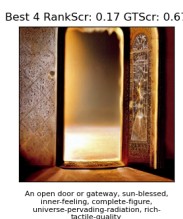

An open door or gateway

An open door or gateway, sun-blessed, auditory-experiences, counterchange-pattern, magic-hour-radiance, worn-appearance

An open door or gateway, sun-blessed, inner-feeling, complete-figure, gentle-sunrise-beams, tiny-openings-on-skin-surface

An open door or gateway, sun-blessed, auditory-experiences, birds-eye-view, gentle-sunrise-beams, high-acutance

An open door or gateway, sun-blessed, inner-feeling, complete-figure, universe-pervading-radiation, rich-tactile-quality

(a) ED-PBRL (Design) - Top Prompts for Sunny GT

**Top Generated Prompts**

| Base Prompt | Best 1 RankScr: 0.05 GTScr: 0.43 | Best 2 RankScr: 0.05 GTScr: 0.45 | Best 3 RankScr: 0.05 GTScr: 0.43 | Best 4 RankScr: 0.05 GTScr: 0.41 |

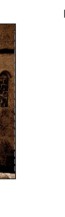 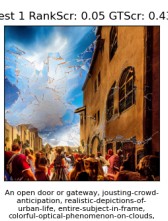 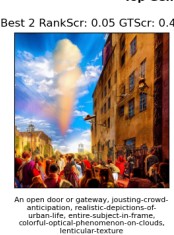 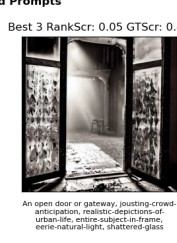 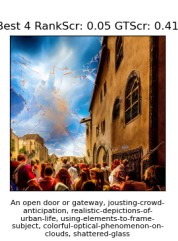

An open door or gateway

An open door or gateway, jousting-crowd-anticipation, realistic-depictions-of-urban-life, entire-subject-in-frame, colorful-optical-phenomenon-on-clouds, shattered-glass

An open door or gateway, jousting-crowd-anticipation, realistic-depictions-of-urban-life, entire-subject-in-frame, colorful-optical-phenomenon-on-clouds, lenticular-texture

An open door or gateway, jousting-crowd-anticipation, realistic-depictions-of-urban-life, entire-subject-in-frame, eerie-natural-light, shattered-glass

An open door or gateway, jousting-crowd-anticipation, realistic-depictions-of-urban-life, using-elements-to-frame-subject, colorful-optical-phenomenon-on-clouds, shattered-glass

(b) Random Exploration - Top Prompts for Sunny GT

Figure 6: Full summary of top generated prompts for the Sunny GT Model. The images compare prompts generated via ED-PBRL (Design) and Random exploration. Each personalized image is annotated with its estimated score from the learned model (RankScore) and its true score from the ground truth model (GTScore), where a higher GTScore indicates better alignment with the target 'Sunny' aesthetic. Note that ED-PBRL consistently finds prompts that yield higher GT Scores, demonstrating its superior personalization capability.

**Top Generated Prompts**

| Base Prompt | Best 1 RankScr: 0.09 GTScr: 0.55 | Best 2 RankScr: 0.09 GTScr: 0.58 | Best 3 RankScr: 0.09 GTScr: 0.59 | Best 4 RankScr: 0.09 GTScr: 0.59 |

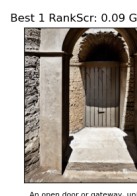 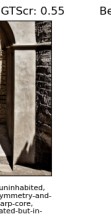 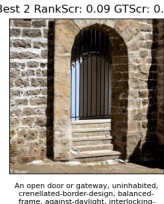 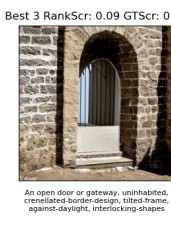 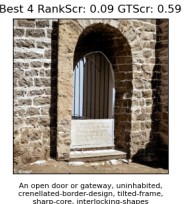

An open door or gateway

An open door or gateway, uninhabited, crenellated-border-design, symmetry-and-asymmetry-balance, sharp-core, components-shown-separated-but-in-relation

An open door or gateway, uninhabited, crenellated-border-design, balanced-frame, against-daylight, interlocking-shapes

An open door or gateway, uninhabited, crenellated-border-design, tilted-frame, against-daylight, interlocking-shapes

An open door or gateway, uninhabited, crenellated-border-design, tilted-frame, sharp-core, interlocking-shapes

(a) ED-PBRL (Design) - Top Prompts for Medieval GT

**Top Generated Prompts**

| Base Prompt | Best 1 RankScr: 0.04 GTScr: 0.10 | Best 2 RankScr: 0.04 GTScr: 0.27 | Best 3 RankScr: 0.04 GTScr: 0.27 | Best 4 RankScr: 0.04 GTScr: 0.16 |

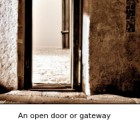 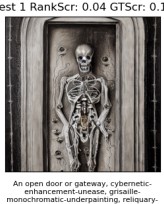 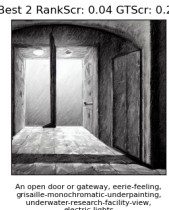 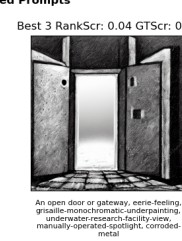 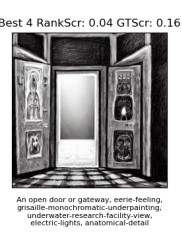

An open door or gateway

An open door or gateway, cybernetic-enhancement-unease, grisaille-monochromatic-underpainting, reliquary-casket-design, amber-hue, anatomical-detail

An open door or gateway, eerie-feeling, grisaille-monochromatic-underpainting, underwater-research-facility-view, electric-lights

An open door or gateway, eerie-feeling, grisaille-monochromatic-underpainting, underwater-research-facility-view, manually-operated-spotlight, corroded-metal

An open door or gateway, eerie-feeling, grisaille-monochromatic-underpainting, underwater-research-facility-view, electric-lights, anatomical-detail

(b) Random Exploration - Top Prompts for Medieval GT

Figure 7: Full summary of top generated prompts for the Medieval GT Model. The images compare prompts generated via ED-PBRL (Design) and Random exploration. Each personalized image is annotated with its estimated score from the learned model (RankScore) and its true score from the ground truth model (GTScore), where a higher GTScore indicates better alignment with the target 'Medieval' aesthetic. Note that ED-PBRL consistently finds prompts that yield higher GT Scores, demonstrating its superior personalization capability.

(a) ED-PBRL (Design) - Top Prompts for Technological GT

(b) Random Exploration - Top Prompts for Technological GT

Figure 8: Full summary of top generated prompts for the Technological GT Model. The images compare prompts generated via ED-PBRL (Design) and Random exploration. Each personalized image is annotated with its estimated score from the learned model (RankScore) and its true score from the ground truth model (GTScore), where a higher GTScore indicates better alignment with the target 'Technological' aesthetic. Note that ED-PBRL consistently finds prompts that yield higher GT Scores, demonstrating its superior personalization capability.

**Qualitative Results from Preliminary Human Study** To illustrate how learned preference models generalize to new prompts, we present qualitative results from a preliminary human study. A participant whose stated preference was for "foresty images with a lot of green, nature and landscapes" provided feedback during the exploration phase. After training, we tested the learned models ($\hat{\theta}_{\text{ED-PBRL}}$ and $\hat{\theta}_{\text{Random}}$) on four new base prompts chosen by the participant. The following figures show the top-ranked personalized images generated by each model.

**Top Generated Prompts**

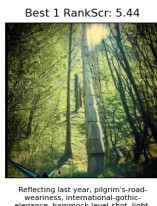 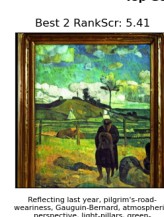 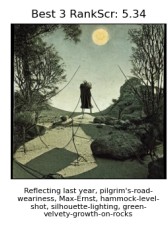 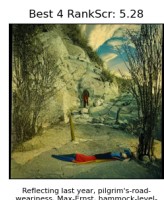

(a) ED-PBRL (Design) - Top Prompts for "Reflecting last year"

**Top Generated Prompts**

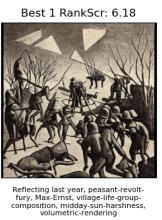 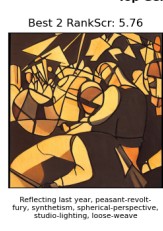 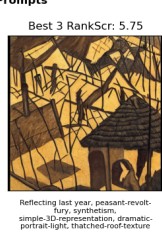 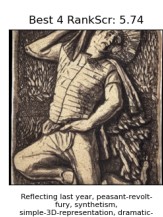

(b) Random Exploration - Top Prompts for "Reflecting last year"

Figure 9: Top generated prompts for the base prompt "Reflecting last year" from the preliminary human study. Images are ranked by the score from the respective learned models (RankScore).

1080

**Top Generated Prompts**

| Base Prompt | Best 1 RankScr: 5.86 | Best 2 RankScr: 5.85 | Best 3 RankScr: 5.84 | Best 4 RankScr: 5.84 |
|---|---|---|---|---|
| 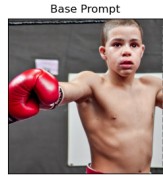 | 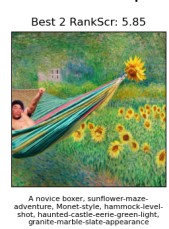 | 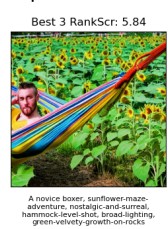 | 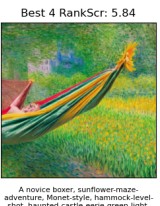 | |

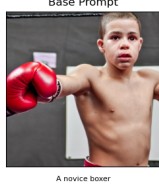

A novice boxer

A novice boxer, sunflower-maze-adventure, Monet-style, hammock-level-shot, misty-beams, green-velvety-growth-on-rocks

A novice boxer, sunflower-maze-adventure, Monet-style, hammock-level-shot, haunted-castle-eerie-green-light, granite-marble-slate-appearance

A novice boxer, sunflower-maze-adventure, nostalgic-and-surreal, hammock-level-shot, broad-lighting, green-velvety-growth-on-rocks

A novice boxer, sunflower-maze-adventure, Monet-style, hammock-level-shot, haunted-castle-eerie-green-light, damp-shaded-environment-indicator

1090
1091

(a) ED-PBRL (Design) - Top Prompts for "A novice boxer"

**Top Generated Prompts**

| Base Prompt | Best 1 RankScr: 5.93 | Best 2 RankScr: 5.93 | Best 3 RankScr: 5.82 | Best 4 RankScr: 5.81 |
|---|---|---|---|---|

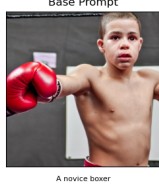 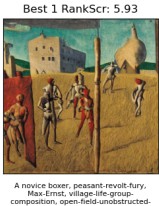 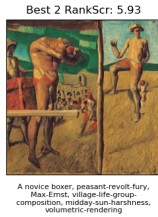 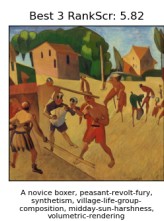 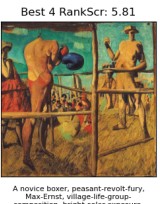

A novice boxer

A novice boxer, peasant-revolt-fury, Max-Ernst, village-life-group-composition, open-field-unobstructed-sunlight, volumetric-rendering

A novice boxer, peasant-revolt-fury, Max-Ernst, village-life-group-composition, midday-sun-harshness, volumetric-rendering

A novice boxer, peasant-revolt-fury, synthetism, village-life-group-composition, midday-sun-harshness, volumetric-rendering

A novice boxer, peasant-revolt-fury, Max-Ernst, village-life-group-composition, bright-solar-exposure, volumetric-rendering

1102
1103

(b) Random Exploration - Top Prompts for "A novice boxer"

1104
1105
1106
1107
1108

Figure 10: Top generated prompts for the base prompt "A novice boxer" from the preliminary human study.

**Top Generated Prompts**

| Base Prompt | Best 1 RankScr: 5.63 | Best 2 RankScr: 5.52 | Best 3 RankScr: 5.50 | Best 4 RankScr: 5.39 |
|---|---|---|---|---|

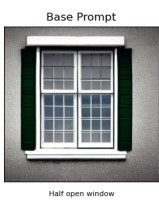 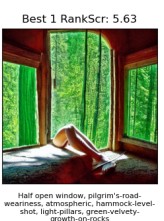 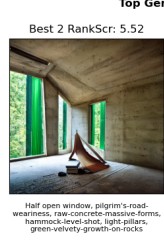 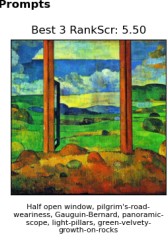 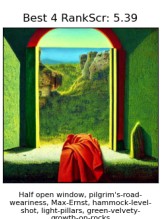

Half open window

Half open window, pilgrim's-road-weariness, atmospheric, hammock-level-shot, light-pillars, green-velvety-growth-on-rocks

Half open window, pilgrim's-road-weariness, raw-concrete-massive-forms, hammock-level-shot, light-pillars, green-velvety-growth-on-rocks

Half open window, pilgrim's-road-weariness, Gauguin-Bernard, panoramic-scope, light-pillars, green-velvety-growth-on-rocks

Half open window, pilgrim's-road-weariness, Max-Ernst, hammock-level-shot, light-pillars, green-velvety-growth-on-rocks

1118
1119

(a) ED-PBRL (Design) - Top Prompts for "Half open window"

**Top Generated Prompts**

| Base Prompt | Best 1 RankScr: 5.90 | Best 2 RankScr: 5.65 | Best 3 RankScr: 5.57 | Best 4 RankScr: 5.51 |
|---|---|---|---|---|

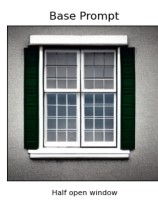 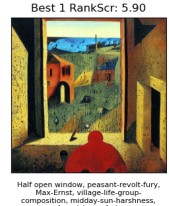 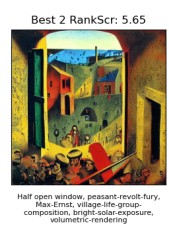 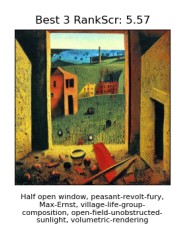 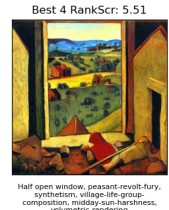

Half open window

Half open window, peasant-revolt-fury, Max-Ernst, village-life-group-composition, midday-sun-harshness, volumetric-rendering

Half open window, peasant-revolt-fury, Max-Ernst, village-life-group-composition, bright-solar-exposure, volumetric-rendering

Half open window, peasant-revolt-fury, Max-Ernst, village-life-group-composition, open-field-unobstructed-sunlight, volumetric-rendering

Half open window, peasant-revolt-fury, synthetism, village-life-group-composition, midday-sun-harshness, volumetric-rendering

1129
1130
1131

(b) Random Exploration - Top Prompts for "Half open window"

1132
1133

Figure 11: Top generated prompts for the base prompt "Half open window" from the preliminary human study.

**Top Generated Prompts**

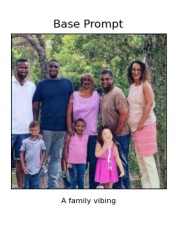 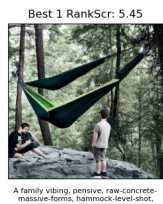 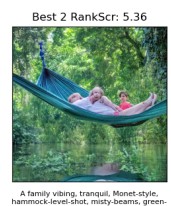 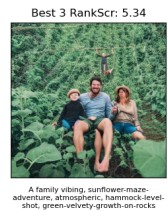 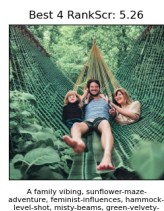

(a) ED-PBRL (Design) - Top Prompts for "A family vibing"

**Top Generated Prompts**

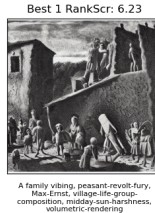 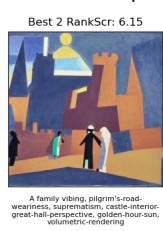 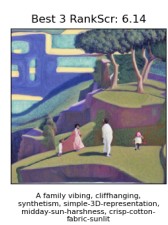 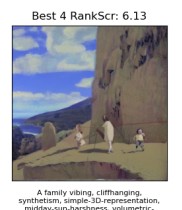

(b) Random Exploration - Top Prompts for "A family vibing"

Figure 12: Top generated prompts for the base prompt "A family vibing" from the preliminary human study.

**Per-Style Panels (All Ten Styles)**   Figure 13 shows per-style held-out accuracy curves for all ten styles (futuristic, sunny, forest, landscape, ancient, noir, watercolor, cyberpunk, minimalist, medieval) at $\lambda = 100$. Each subplot reports mean held-out accuracy across 3 cross-validation folds for ED-PBRL and Random; curves are shown at training sizes 10/20/30/40/50 episodes with 10 episodes held out for testing.

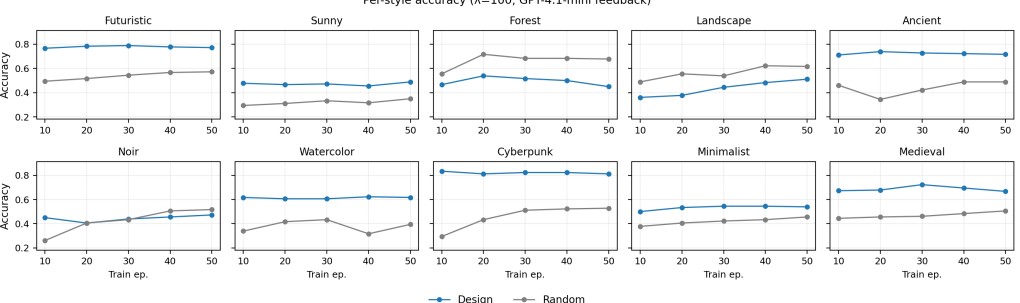

Figure 13: LLM-simulated preference study, all ten styles, $\lambda = 100$: per-style held-out accuracy (10 test episodes). ED-PBRL consistently outperforms random exploration across most styles, with the advantage being most pronounced when training data is limited.

**Comprehensive Results Table**   Table 2 lists per-style held-out accuracies at $\lambda = 100$ for both ED-PBRL (Design) and Random exploration, their differences in percentage points, and the number of hold-out comparisons. Results are shown for 50 training episodes with 10 held-out test episodes, averaged across 3 cross-validation folds.

Table 2: LLM-simulated preference held-out accuracy by style at $\lambda = 100$ and 50 training episodes. Accuracy values are in percentage points; Diff is Design minus Random. Hold-out consists of 10 test episodes (60 comparisons), averaged across 3 cross-validation folds.

| Style | Design (%) | Random (%) | Diff (pp) |
|---|---|---|---|
| Futuristic | 77.2 | 57.2 | +20.0 |
| Sunny | 48.9 | 35.0 | +13.9 |
| Forest | 45.0 | 67.8 | −22.8 |
| Landscape | 51.1 | 61.7 | −10.6 |
| Ancient | 71.7 | 48.9 | +22.8 |
| Noir | 47.2 | 51.7 | −4.4 |
| Watercolor | 61.7 | 39.4 | +22.2 |
| Cyberpunk | 81.1 | 52.8 | +28.3 |
| Minimalist | 53.9 | 45.6 | +8.3 |
| Medieval | 66.7 | 50.6 | +16.1 |
| **Average** | **60.4** | **51.1** | **+9.4** |

# B APPENDIX: PROOFS, DERIVATIONS AND FURTHER RESULTS

## B.1 RELATIONSHIP BETWEEN ESTIMATION ERROR AND FISHER INFORMATION

We formalize the link between estimation error and information used in the main text. For the regularized MLE $\theta_\lambda$, we show that the Mean Squared Error (MSE) matrix is controlled by the inverse of the regularized Fisher Information at the true parameter $\theta^*$:

$$\mathbb{E}\left[(\theta_\lambda - \theta^*)(\theta_\lambda - \theta^*)^\top\right] \preceq C_{\theta^*}^\lambda \cdot I_\lambda(\theta^*)^{-1}.$$

The proof relies on a self-concordant analysis of the (regularized) log-likelihood to relate the random Hessian to $I_\lambda(\theta^*)$ within a local neighborhood of $\theta^*$. We state the precise assumptions next and then provide the full proof.

### B.1.1 ASSUMPTIONS FOR THE MSE BOUND

The derivation of the bound relies on two key assumptions. We state them formally here before proceeding with the proof.

**Assumption B.1** (Uniform Local Consistency). *For a given experimental design (policies, $T$, $\lambda$), we assume the regularized MLE $\theta_\lambda$ is close to the true parameter $\theta^*$. Specifically, we assume there exists a constant $r_{\theta^*}^\lambda \in [0, 1)$, dependent on the problem parameters but not on the random data realization, that uniformly bounds the distance between $\theta_\lambda$ and $\theta^*$ in the local norm defined by the FIM at $\theta^*$:*

$$\|\theta_\lambda - \theta^*\|_{I(\theta^*)} \equiv \sqrt{(\theta_\lambda - \theta^*)^T I(\theta^*)(\theta_\lambda - \theta^*)} \leq r_{\theta^*}^\lambda$$

*This assumption is a prerequisite for our finite-sample analysis. It allows us to define a data-independent constant $C_{\theta^*}^\lambda = (1 - r_{\theta^*}^\lambda)^{-4}$ that can be moved outside the expectation in the proof, simplifying the analysis. Standard large-sample theory for MLE suggests that for a sufficiently large number of samples, this condition is expected to hold with high probability.*

**Assumption B.2** (Bounded True Parameter). *The squared $\ell_2$-norm of the true parameter vector $\theta^*$ is bounded relative to the regularization strength $\lambda$:*

$$\|\theta^*\|_2^2 \leq \frac{1}{\lambda}$$

*This assumption, which is standard in the analysis of ridge regression and regularized estimators Mutný (2024), constrains the magnitude of the true parameter relative to the regularization strength. It ensures that the bias introduced by the $\ell_2$ penalty does not overwhelm the information-related terms in the analysis. In matrix terms, this implies $\lambda^2 \theta^* (\theta^*)^T \preceq \lambda I_d$.*

With these conditions explicitly stated, we can now present the formal theorem and its proof.

**Theorem B.1** (Upper Bound on MSE). *Let $\theta_\lambda$ be the regularized maximum likelihood estimator and $\theta^*$ be the true parameter. Under Assumptions B.1 and B.2, the Mean Squared Error (MSE) matrix of the estimator is bounded by:*

$$\mathbb{E}[(\theta_\lambda - \theta^*)(\theta_\lambda - \theta^*)^T] \preceq C_{\theta^*}^\lambda \cdot I_\lambda(\theta^*)^{-1}$$

*where $I_\lambda(\theta^*) = I(\theta^*) + \lambda I_d$ is the regularized Fisher Information Matrix at the true parameter, and $C_{\theta^*}^\lambda = (1 - r_{\theta^*}^\lambda)^{-4}$ is a constant determined by the radius $r_{\theta^*}^\lambda$ from Assumption B.1.*

*Proof.* The proof proceeds in three main steps. First, we establish an exact expression for the estimation error using a Taylor expansion. Second, we use the self-concordance property of the negative log-likelihood, combined with Assumption B.1, to bound the random Hessian that appears in the error expression. Finally, we combine these results and use Assumption B.2 to derive the upper bound on the MSE.

**Step 1: Taylor Expansion of the Score Function.** The estimator $\theta_\lambda$ is the solution to the regularized maximum likelihood problem, defined by the first-order optimality condition $s_\lambda(\theta_\lambda) = 0$. The regularized score function $s_\lambda(\theta)$ is the gradient of the regularized log-likelihood:

$$s_\lambda(\theta) = \nabla L_{reg}(\theta) = \sum_{t=1}^{T} \sum_{q=1}^{K} (y_{t,q} - p_{t,q}(\theta))x_{t,q} - \lambda\theta$$

The Hessian of the negative regularized log-likelihood is the regularized Fisher Information Matrix, $I_\lambda(\theta) = -\nabla^2 L_{reg}(\theta) = I(\theta) + \lambda I_d$. Note that $\nabla s_\lambda(\theta) = -I_\lambda(\theta)$.

By the vector-valued Mean Value Theorem (a form of Taylor's theorem), we can expand the function $s_\lambda(\theta)$ around the true parameter $\theta^*$:

$$0 = s_\lambda(\theta_\lambda) = s_\lambda(\theta^*) + \nabla s_\lambda(\theta_\tau)(\theta_\lambda - \theta^*)$$

for some $\theta_\tau$ on the line segment between $\theta^*$ and $\theta_\lambda$. Substituting $\nabla s_\lambda(\theta_\tau) = -I_\lambda(\theta_\tau)$, we get:

$$0 = s_\lambda(\theta^*) - I_\lambda(\theta_\tau)(\theta_\lambda - \theta^*)$$

Rearranging gives the exact expression for the estimation error:

$$\theta_\lambda - \theta^* = I_\lambda(\theta_\tau)^{-1} s_\lambda(\theta^*)$$

The MSE matrix is therefore given by the expectation:

$$\mathbb{E}[(\theta_\lambda - \theta^*)(\theta_\lambda - \theta^*)^T] = \mathbb{E}[I_\lambda(\theta_\tau)^{-1} s_\lambda(\theta^*) s_\lambda(\theta^*)^\top I_\lambda(\theta_\tau)^{-1}]$$

**Step 2: Bounding the Hessian via Self-Concordance.** The main difficulty is relating the terms $I_\lambda(\theta_\tau)$ and $s_\lambda(\theta^*)$ in the error expression, as they are evaluated at different points. We resolve this by bounding the Hessian term $I_\lambda(\theta_\tau)$ using the self-concordance property of the unregularized negative log-likelihood function, $L(\theta) = -\log P(\text{data}|\theta)$.

The negative log-likelihood for multinomial logistic regression is a sum of log-sum-exp functions, which is a standard example of a self-concordant function. Its Hessian is the Fisher Information Matrix, $I(\theta) = \nabla^2 L(\theta)$. For a self-concordant function $f$, the Hessians at two points $x, y$ are related by $(1 - \|y-x\|_x)^2 \nabla^2 f(x) \preceq \nabla^2 f(y)$ provided that the local norm $\|y-x\|_x = \sqrt{(y-x)^T \nabla^2 f(x)(y-x)}$ is less than 1.

We now invoke **Assumption B.1**, which states that $\|\theta_\lambda - \theta^*\|_{I(\theta^*)} \leq r_{\theta^*}^\lambda < 1$ for all data realizations. Since $\theta_\tau$ lies on the line segment between $\theta^*$ and $\theta_\lambda$, it is necessarily closer to $\theta^*$ in any norm, including the local norm defined by $I(\theta^*)$. Thus, $\|\theta_\tau - \theta^*\|_{I(\theta^*)} \leq \|\theta_\lambda - \theta^*\|_{I(\theta^*)} \leq r_{\theta^*}^\lambda$.

Applying the self-concordance property with $f(\theta) = L(\theta)$, $x = \theta^*$, and $y = \theta_\tau$, we get a lower bound on the unregularized FIM:

$$I(\theta_\tau) \succeq (1 - \|\theta_\tau - \theta^*\|_{I(\theta^*)})^2 I(\theta^*) \succeq (1 - r_{\theta^*}^\lambda)^2 I(\theta^*)$$

This inequality holds deterministically for any realization of the data due to our assumption. We use this to bound the regularized FIM:

$$I_\lambda(\theta_\tau) = I(\theta_\tau) + \lambda I_d$$
$$\succeq (1 - r_{\theta^*}^\lambda)^2 I(\theta^*) + \lambda I_d$$
$$\succeq (1 - r_{\theta^*}^\lambda)^2 I(\theta^*) + (1 - r_{\theta^*}^\lambda)^2 \lambda I_d \quad \text{(since } 0 < (1 - r_{\theta^*}^\lambda)^2 \leq 1 \text{ and } \lambda I_d \text{ is pos. semidef.)}$$
$$= (1 - r_{\theta^*}^\lambda)^2 (I(\theta^*) + \lambda I_d) = (1 - r_{\theta^*}^\lambda)^2 I_\lambda(\theta^*)$$

Inverting this matrix inequality (using the property that if $A \succeq B \succ 0$, then $B^{-1} \succeq A^{-1} \succ 0$) yields an upper bound on the inverse Hessian:

$$I_\lambda(\theta_\tau)^{-1} \preceq (1 - r_{\theta^*}^\lambda)^{-2} I_\lambda(\theta^*)^{-1}$$

**Step 3: Deriving the Final MSE Bound.** We substitute the bound on the inverse Hessian back into the MSE expression. Since the bound holds deterministically for a constant $r_{\theta^*}^\lambda$, the term $(1 - r_{\theta^*}^\lambda)^{-2}$ is a constant and can be manipulated outside the expectation.

$$
\begin{aligned}
\mathbb{E}[(\theta_\lambda - \theta^*)(\theta_\lambda - \theta^*)^T] &= \mathbb{E}[I_\lambda(\theta_\tau)^{-1} s_\lambda(\theta^*) s_\lambda(\theta^*)^\top I_\lambda(\theta_\tau)^{-1}] \\
&\preceq \mathbb{E}\left[\left((1 - r_{\theta^*}^\lambda)^{-2} I_\lambda(\theta^*)^{-1}\right) s_\lambda(\theta^*) s_\lambda(\theta^*)^\top \left((1 - r_{\theta^*}^\lambda)^{-2} I_\lambda(\theta^*)^{-1}\right)\right] \\
&= (1 - r_{\theta^*}^\lambda)^{-4} I_\lambda(\theta^*)^{-1} \mathbb{E}[s_\lambda(\theta^*) s_\lambda(\theta^*)^\top] I_\lambda(\theta^*)^{-1}
\end{aligned}
$$

Next, we analyze the expectation of the outer product of the regularized score at the true parameter, $\mathbb{E}[s_\lambda(\theta^*) s_\lambda(\theta^*)^\top]$. Let $s(\theta^*)$ be the unregularized score. We know that $\mathbb{E}[s(\theta^*)] = 0$ and $\mathbb{E}[s(\theta^*) s(\theta^*)^\top] = I(\theta^*)$ (by the Information Matrix Equality).

$$
\begin{aligned}
\mathbb{E}[s_\lambda(\theta^*) s_\lambda(\theta^*)^\top] &= \mathbb{E}[(s(\theta^*) - \lambda \theta^*)(s(\theta^*) - \lambda \theta^*)^\top] \\
&= \mathbb{E}[s(\theta^*) s(\theta^*)^\top] - \lambda \mathbb{E}[s(\theta^*)](\theta^*)^\top - \lambda \theta^* \mathbb{E}[s(\theta^*)^\top] + \lambda^2 \theta^* (\theta^*)^\top \\
&= I(\theta^*) - 0 - 0 + \lambda^2 \theta^* (\theta^*)^\top \\
&= I(\theta^*) + \lambda^2 \theta^* (\theta^*)^\top
\end{aligned}
$$

Now, we invoke **Assumption B.2**, which states $\|\theta^*\|_2^2 \leq 1/\lambda$. This implies that $\lambda^2 \theta^* (\theta^*)^T \preceq \lambda I_d$. Using this, we can bound the expected score term:

$$\mathbb{E}[s_\lambda(\theta^*) s_\lambda(\theta^*)^\top] = I(\theta^*) + \lambda^2 \theta^* (\theta^*)^\top \preceq I(\theta^*) + \lambda I_d = I_\lambda(\theta^*)$$

Finally, substituting this back into the MSE expression gives the result:

$$
\begin{aligned}
\mathbb{E}[(\theta_\lambda - \theta^*)(\theta_\lambda - \theta^*)^T] &\preceq (1 - r_{\theta^*}^\lambda)^{-4} I_\lambda(\theta^*)^{-1} \left(\mathbb{E}[s_\lambda(\theta^*) s_\lambda(\theta^*)^\top]\right) I_\lambda(\theta^*)^{-1} \\
&\preceq (1 - r_{\theta^*}^\lambda)^{-4} I_\lambda(\theta^*)^{-1} I_\lambda(\theta^*) I_\lambda(\theta^*)^{-1} \\
&= \frac{1}{(1 - r_{\theta^*}^\lambda)^4} I_\lambda(\theta^*)^{-1}
\end{aligned}
$$

This establishes the bound with constant $C_{\theta^*}^\lambda = (1 - r_{\theta^*}^\lambda)^{-4}$, concluding the proof. □

### B.2 DERIVATION OF THE FISHER INFORMATION MATRIX FOR MULTINOMIAL LOGISTIC REGRESSION

The Fisher Information Matrix (FIM) quantifies the amount of information that an observable random variable carries about an unknown parameter $\theta$ upon which the probability of the random variable depends. Here, we derive the FIM for a single preference observation within a multinomial logistic regression model.

Consider a single observation where an expert chooses one item from a set of $K$ items. Let $\mathbf{x}_q \in \mathbb{R}^d$ be the feature vector associated with item $q \in \{1, \ldots, K\}$. The probability of the expert choosing item $q$, given the parameter vector $\theta \in \mathbb{R}^d$, is modeled by the softmax function:

$$p_q(\theta) = P(\text{item } q \text{ is chosen}|\mathbf{x}_1, \ldots, \mathbf{x}_K, \theta) = \frac{\exp(\theta^\top \mathbf{x}_q)}{\sum_{q'=1}^K \exp(\theta^\top \mathbf{x}_{q'})}$$

Let $y_q$ be an indicator variable such that $y_q = 1$ if item $q$ is chosen, and $y_q = 0$ otherwise. Note that $\sum_{q=1}^K y_q = 1$. The log-likelihood for this single observation is:

$$\mathcal{L}(\theta) = \sum_{q=1}^K y_q \log p_q(\theta)$$

The score vector, which is the gradient of the log-likelihood with respect to $\theta$, is:

$$S(\theta) = \nabla_\theta \mathcal{L}(\theta) = \sum_{q=1}^K y_q \frac{1}{p_q(\theta)} \nabla_\theta p_q(\theta)$$

The gradient of $p_q(\theta)$ is $\nabla_\theta p_q(\theta) = p_q(\theta)(\mathbf{x}_q - \bar{\mathbf{x}}(\theta))$, where $\bar{\mathbf{x}}(\theta) = \sum_{q'=1}^{K} p_{q'}(\theta)\mathbf{x}_{q'}$ is the expected feature vector under the current model. Substituting this into the score function:

$$S(\theta) = \sum_{q=1}^{K} y_q(\mathbf{x}_q - \bar{\mathbf{x}}(\theta)) = \left(\sum_{q=1}^{K} y_q\mathbf{x}_q\right) - \bar{\mathbf{x}}(\theta)$$

This can also be written as $S(\theta) = \sum_{q=1}^{K}(y_q - p_q(\theta))\mathbf{x}_q$.

The Hessian matrix $H(\theta)$ is the matrix of second derivatives of the log-likelihood: $H(\theta) = \nabla_\theta S(\theta)^\top$.

$$H(\theta) = \nabla_\theta \left(\sum_{q=1}^{K} y_q\mathbf{x}_q - \sum_{q'=1}^{K} p_{q'}(\theta)\mathbf{x}_{q'}\right)^\top = -\nabla_\theta \left(\sum_{q'=1}^{K} p_{q'}(\theta)\mathbf{x}_{q'}\right)^\top$$

Calculating the derivative:

$$\nabla_\theta \left(\sum_{q'=1}^{K} p_{q'}(\theta)\mathbf{x}_{q'}\right)^\top = \sum_{q'=1}^{K} \left((\nabla_\theta p_{q'}(\theta))\mathbf{x}_{q'}^\top + p_{q'}(\theta)\nabla_\theta \mathbf{x}_{q'}^\top\right)$$

$$= \sum_{q'=1}^{K} p_{q'}(\theta)(\mathbf{x}_{q'} - \bar{\mathbf{x}}(\theta))\mathbf{x}_{q'}^\top \quad (\text{since } \nabla_\theta \mathbf{x}_{q'}^\top = 0)$$

$$= \sum_{q'=1}^{K} p_{q'}(\theta)\mathbf{x}_{q'}\mathbf{x}_{q'}^\top - \left(\sum_{q'=1}^{K} p_{q'}(\theta)\mathbf{x}_{q'}\right)\left(\sum_{q''=1}^{K} p_{q''}(\theta)\mathbf{x}_{q''}\right)^\top$$

$$= \sum_{q'=1}^{K} p_{q'}(\theta)\mathbf{x}_{q'}\mathbf{x}_{q'}^\top - \bar{\mathbf{x}}(\theta)\bar{\mathbf{x}}(\theta)^\top$$

So, the Hessian is:

$$H(\theta) = -\left(\sum_{q'=1}^{K} p_{q'}(\theta)\mathbf{x}_{q'}\mathbf{x}_{q'}^\top - \bar{\mathbf{x}}(\theta)\bar{\mathbf{x}}(\theta)^\top\right)$$

The Fisher Information Matrix $I(\theta)$ for this single observation is defined as the negative expectation of the Hessian: $I(\theta) = -\mathbb{E}[H(\theta)]$. Since the Hessian $H(\theta)$ as derived here does not depend on the random outcome variables $y_q$ (after simplification using properties of $p_q(\theta)$), the expectation does not change it. Thus:

$$I(\theta) = \sum_{q'=1}^{K} p_{q'}(\theta)\mathbf{x}_{q'}\mathbf{x}_{q'}^\top - \bar{\mathbf{x}}(\theta)\bar{\mathbf{x}}(\theta)^\top$$

Expanding $\bar{\mathbf{x}}(\theta) = \sum_{q=1}^{K} p_q(\theta)\mathbf{x}_q$, the term $\bar{\mathbf{x}}(\theta)\bar{\mathbf{x}}(\theta)^\top$ becomes $\left(\sum_{q=1}^{K} p_q(\theta)\mathbf{x}_q\right)\left(\sum_{q'=1}^{K} p_{q'}(\theta)\mathbf{x}_{q'}\right)^\top = \sum_{q,q'} p_q(\theta)p_{q'}(\theta)\mathbf{x}_q\mathbf{x}_{q'}^\top$. Thus, we get the form:

$$I(\theta) = \sum_{q=1}^{K} p_q(\theta)\mathbf{x}_q\mathbf{x}_q^\top - \sum_{q,q'} p_q(\theta)p_{q'}(\theta)\mathbf{x}_q\mathbf{x}_{q'}^\top$$

This expression represents the FIM for one multinomial preference choice. If there are $N$ independent such choices, the total FIM is the sum of the FIMs from each choice. This derivation provides the basis for the FIM expressions used in the subsequent experimental design.

### B.3 EXPECTED FISHER INFORMATION OBJECTIVE FOR PBRL

Our goal is to select $K$ policies, $\pi_{1:K} = (\pi_1, \ldots, \pi_K)$, to maximize information about the unknown parameter $\theta$. A classical challenge in Optimal Experimental Design (OED) is that directly optimizing a discrete set of experiments (trajectories in our case) is often intractable Pukelsheim (2006); Fedorov

& Hackl (1997). A standard approach in OED is to instead optimize a design measure, which in our policy-based setting corresponds to optimizing over policies and considering the *expected* Fisher Information Matrix (FIM) they induce.

The total expected regularized FIM, $I_{reg}(\pi_{1:K}, \theta)$, for $K$ policies $\pi_{1:K}$ generating $T$ episodes of $H$ steps each is:

$$I_{reg}(\pi_{1:K}, \theta) = T \sum_{h=1}^{H} I_h(\pi_{1:K}, \theta) + \lambda I_d$$

Here, $I_h(\pi_{1:K}, \theta)$ is the expected FIM contribution from timestep $h$ of a single episode, averaged over the trajectory distributions $\eta_{\pi_q}$ induced by each policy $\pi_q$. Let $s_h^q$ be the state of trajectory $\tau_q \sim \eta_{\pi_q}$ at step $h$, and $p(q|h; \tau_{1..K})$ be the softmax probability of preferring state $s_h^q$ from the set of $K$ states $\{s_h^1, \ldots, s_h^K\}$ presented at that step. Then $I_h(\pi_{1:K}, \theta)$ is: $I_h(\pi_{1:K}, \theta) =$
$$\mathbb{E}_{\substack{\tau_q \sim \eta_{\pi_q} \\ q \in [K]}} \left[ \sum_{q=1}^{K} p(q|h; \tau_{1..K}) \phi(s_h^q) \phi(s_h^q)^\top - \sum_{q,q'} p(q|h; \tau_{1..K}) p(q'|h; \tau_{1..K}) \phi(s_h^q) \phi(s_h^{q'})^\top \right]$$

The detailed FIM derivation for a single multinomial choice is in Appendix B.2.

The ideal experimental design objective is to choose policies $\pi_{1:K}$ to optimize a scalar criterion $s(\cdot)$ of this expected FIM (e.g., D- or A-optimality):

$$\arg\max_{\pi_{1:K}} s\left(I_{reg}(\pi_{1:K}, \theta)\right) \tag{7}$$

The challenges associated with this ideal objective are discussed in Section 4, and are addressed by the reformulation and approximation techniques detailed in the main text (Section 4.3) and expanded upon in Section B.4 below.

### B.4 REFORMULATION TO A TRACTABLE OBJECTIVE

This section provides the full derivation of the tractable experimental design objective discussed in Section 4.3, following the three main steps outlined there.

**Step 1: Reformulation using State Visitation Measures.** We begin with the expected regularized FIM from Eq. 7, defined as $I_{reg}(\pi_{1:K}, \theta) = T \sum_{h=1}^{H} I_h(\pi_{1:K}, \theta) + \lambda I_d$. The core of the derivation is to reformulate the per-timestep FIM contribution, $I_h(\pi_{1:K}, \theta)$, in terms of state visitation measures. This reformulation is formalized by the following lemma.

**Lemma B.2.** *Let $\pi_1, \ldots, \pi_K$ be policies with corresponding trajectory distributions $\eta_{\pi_1}, \ldots, \eta_{\pi_K}$ and state visitation measures $d_{\pi_1}^h, \ldots, d_{\pi_K}^h$. Let $f(s_1, \ldots, s_K)$ be any function of a tuple of $K$ states. Assume trajectories $\tau_1, \ldots, \tau_K$ are drawn independently, $\tau_q \sim \eta_{\pi_q}$. Let $s_h^q$ denote the state at timestep $h$ of trajectory $\tau_q$. Then for any fixed $h$:*

$$\mathbb{E}_{\substack{\tau_q \sim \eta_{\pi_q} \\ q \in [K]}} \left[ f(s_h^1, \ldots, s_h^K) \right] = \mathbb{E}_{\substack{s_q \sim d_{\pi_q}^h \\ q \in [K]}} \left[ f(s_1, \ldots, s_K) \right]$$

*where the expectation on the right is taken with respect to states $s_1, \ldots, s_K$ drawn independently from the respective state visitation measures at timestep $h$. The notation $s_q \sim d_{\pi_q}^h$ for $q \in [K]$ implies the joint draw $(s_1, \ldots, s_K)$ is from the product distribution $\prod_{q=1}^{K} d_{\pi_q}^h$.*

*Proof of Lemma B.2.* For a fixed $h$, we have:

$$
\mathop{\mathbb{E}}_{\substack{\tau_q \sim \eta_{\pi_q} \\ q \in [K]}} \left[ f(s_h^1, \ldots, s_h^K) \right] = \sum_{\tau_1, \ldots, \tau_K \in \mathcal{T}} \left( \prod_{q=1}^K \eta_{\pi_q}(\tau_q) \right) f(s_h^1, \ldots, s_h^K)
$$

$$
= \sum_{\tau_1, \ldots, \tau_K \in \mathcal{T}} \left( \prod_{q=1}^K \eta_{\pi_q}(\tau_q) \right) \sum_{s_1, \ldots, s_K \in \mathcal{S}} f(s_1, \ldots, s_K) \prod_{q=1}^K \mathbb{I}_{\{s_q = s_h^q\}} \quad \text{(Introduce indicators)}
$$

$$
= \sum_{s_1, \ldots, s_K \in \mathcal{S}} f(s_1, \ldots, s_K) \sum_{\tau_1, \ldots, \tau_K \in \mathcal{T}} \left( \prod_{q=1}^K \eta_{\pi_q}(\tau_q) \mathbb{I}_{\{s_q = s_h^q\}} \right) \quad \text{(Rearrange sums)}
$$

$$
= \sum_{s_1, \ldots, s_K \in \mathcal{S}} f(s_1, \ldots, s_K) \left( \prod_{q=1}^K \sum_{\tau_q \in \mathcal{T}} \eta_{\pi_q}(\tau_q) \mathbb{I}_{\{s_q = s_h^q\}} \right) \quad \text{(Factorize sum over } \tau)
$$

$$
= \sum_{s_1, \ldots, s_K \in \mathcal{S}} f(s_1, \ldots, s_K) \left( \prod_{q=1}^K d_{\pi_q}^h(s_q) \right) \quad \text{(Definition of } d_{\pi_q}^h)
$$

$$
= \mathop{\mathbb{E}}_{\substack{s_q \sim d_{\pi_q}^h \\ q \in [K]}} \left[ f(s_1, \ldots, s_K) \right] \quad \text{(Definition of expectation w.r.t. product measure)}
$$

This completes the proof. $\qquad\qquad\square$

Using this lemma, the per-timestep expected FIM $I_h(\pi_{1:K}, \theta)$ can be equivalently expressed in terms of state visitation measures. Let $d_{1:K}^h = (d_{\pi_1}^h, \ldots, d_{\pi_K}^h)$. The equality $I_h(\pi_{1:K}, \theta) = I_h(d_{1:K}^h, \theta)$ signifies that the problem is now over the space of visitation measures. However, this expression is still dependent on the unknown $\theta$. In particular, invoking Lemma B.2 with the choice

$$
f(s_1, \ldots, s_K) = \sum_{q=1}^K p(q \mid h; s_{1:K}, \theta) \, \phi(s_q)\phi(s_q)^\top - \Big( \sum_{q=1}^K p(q \mid h; s_{1:K}, \theta) \, \phi(s_q) \Big) \Big( \sum_{q'=1}^K p(q' \mid h; s_{1:K}, \theta) \, \phi(s_{q'}) \Big)^\top,
$$

we obtain the explicit per-step form

$$
\begin{aligned}
I_h(d_{1:K}^h, \theta) = \mathop{\mathbb{E}}_{\substack{s_q \sim d_{\pi_q}^h \\ q \in [K]}} \left[ f(s_1, \ldots, s_K) \right] = \mathop{\mathbb{E}}_{\substack{s_q \sim d_{\pi_q}^h \\ q \in [K]}} & \left[ \sum_{q=1}^K p(q \mid h; s_{1:K}, \theta) \, \phi(s_q)\phi(s_q)^\top \right. \\
& \left. - \Big( \sum_{q=1}^K p(q \mid h; s_{1:K}, \theta) \, \phi(s_q) \Big) \Big( \sum_{q'=1}^K p(q' \mid h; s_{1:K}, \theta) \, \phi(s_{q'}) \Big)^\top \right].
\end{aligned}
\tag{8}
$$

Consequently, defining $I_\lambda(d_{1:K}, \theta) = T \sum_{h=1}^H I_h(d_{1:K}^h, \theta) + \lambda I_d$, the Step 1 design problem reads

$$
\mathop{\arg\max}_{d_{1:K} \in \mathcal{D}_{sv}} \; s(I_\lambda(d_{1:K}, \theta)),
\tag{9}
$$

where $\mathcal{D}_{sv}$ denotes the set of valid collections of visitation measures. This formulation (still) carries the $K$-tuple coupling through $p(\cdot \mid h; s_{1:K}, \theta)$ and depends on the unknown $\theta$, hence it remains computationally challenging and motivates the subsequent steps.

**Step 2: $\theta$-agnostic approximation.** The per-step information $I_h(d_{1:K}^h, \theta)$ depends on $\theta$ only through the softmax probabilities $p(\cdot)$. To design queries without a reliable prior—and avoid brittleness to misspecification—we adopt a symmetric average-case surrogate that is standard in fixed-design OED: replace $p(q \mid h; s_{1:K}, \theta)$ by its expectation under a symmetric, uninformative prior. This yields

$$
p(q \mid s_{1:K}) \approx \frac{1}{K}.
$$

For $K = 2$, this equality holds for any symmetric prior; for $K > 2$, it is a common and reasonable $\theta$-agnostic approximation when the $K$ alternatives are constructed to be diverse/symmetric. It can also be viewed as a homoscedasticity assumption on choice uncertainty when the parameter-dependent

heteroscedasticity is unknown Pukelsheim (2006). A fully Bayesian alternative—computing $\mathbb{E}_\theta[p(q \mid h; s_{1:K}, \theta)]$ for every tuple $s_{1:K}$—is computationally prohibitive at design time.

Substituting the uniform surrogate into $I_h(d_{1:K}^h, \theta)$ yields the approximate expected FIM contribution at timestep $h$, denoted $\tilde{I}_h(d_{1:K}^h)$, which is now independent of $\theta$:

$$\tilde{I}_h(d_{1:K}^h) = \mathop{\mathbb{E}}_{\substack{s_q \sim d_q^h \\ q \in [K]}} \left[ \frac{1}{K} \sum_{q=1}^K \phi(s_q)\phi(s_q)^\top - \frac{1}{K^2} \sum_{q,q'=1}^K \phi(s_q)\phi(s_{q'})^\top \right] \tag{10}$$

We denote this per-timestep quantity by $\tilde{I}_h(d_{1:K}^h)$ throughout, consistent with the main text (cf. Eq. 5). While now $\theta$-independent, computing this expectation naively still involves a sum over $|\mathcal{S}|^K$ terms.

**Step 3: Marginalization for Tractability.** The expectation in Eq. 10 can be marginalized to yield a tractable closed-form expression, as established by the following theorem.

**Theorem B.3.** *Let $d_q^h$ be the state visitation measure for policy $\pi_q$ at step $h$, for $q \in [K]$. Under the uniform preference approximation, the expected FIM contribution $\tilde{I}_h(d_{1:K}^h)$ can be rewritten as:*

$$\tilde{I}_h(d_{1:K}^h) = \frac{1}{K} \sum_{q=1}^K \sum_{s \in \mathcal{S}} d_q^h(s)\phi(s)\phi(s)^\top - \frac{1}{K^2} \sum_{q,q'=1}^K \left( \sum_{s \in \mathcal{S}} d_q^h(s)\phi(s) \right) \left( \sum_{s' \in \mathcal{S}} d_{q'}^h(s')\phi(s')^\top \right)$$

*Furthermore, in matrix notation, where $\Phi \in \mathbb{R}^{|\mathcal{S}| \times d}$ is the feature matrix, $d_q^h \in \mathbb{R}^{|\mathcal{S}|}$ is the state visitation vector, and $\bar{d}^h = \frac{1}{K} \sum_{q=1}^K d_q^h$ is the average visitation vector:*

$$\tilde{I}_h(d_{1:K}^h) = \Phi^T \left( \frac{1}{K} \sum_{q=1}^K \operatorname{diag}(d_q^h) - \bar{d}^h(\bar{d}^h)^T \right) \Phi \tag{11}$$

*Proof of Theorem B.3.* We start with the definition of $\tilde{I}_h(d_{1:K}^h)$ under the uniform approximation from Eq. 10. Let $A_h = \mathbb{E}\left[ \frac{1}{K} \sum_{q=1}^K \phi(s_q)\phi(s_q)^\top \right]$ and $B_h = \mathbb{E}\left[ \frac{1}{K^2} \sum_{q,q'=1}^K \phi(s_q)\phi(s_{q'})^\top \right]$. By linearity of expectation:

$$A_h = \frac{1}{K} \sum_{q=1}^K \mathbb{E}[\phi(s_q)\phi(s_q)^\top] = \frac{1}{K} \sum_{q=1}^K \sum_{s \in \mathcal{S}} d_q^h(s)\phi(s)\phi(s)^\top.$$

For $B_h$, since $s_q$ and $s_{q'}$ are independent for $q \neq q'$:

$$B_h = \frac{1}{K^2} \sum_{q,q'=1}^K \mathbb{E}[\phi(s_q)\phi(s_{q'})^\top] = \frac{1}{K^2} \sum_{q,q'=1}^K \mathbb{E}[\phi(s_q)]\,\mathbb{E}[\phi(s_{q'})]^\top$$

$$= \frac{1}{K^2} \sum_{q,q'=1}^K \left( \sum_{s \in \mathcal{S}} d_q^h(s)\phi(s) \right) \left( \sum_{s' \in \mathcal{S}} d_{q'}^h(s')\phi(s')^\top \right).$$

This yields the first result. For the matrix form, we use the fact that $\sum_s d_q^h(s)\phi(s)\phi(s)^\top = \Phi^T \operatorname{diag}(d_q^h)\Phi$ and $\sum_s d_q^h(s)\phi(s) = \Phi^T d_q^h$.

$$A_h = \frac{1}{K} \sum_{q=1}^K \Phi^T \operatorname{diag}(d_q^h)\Phi = \Phi^T \left( \frac{1}{K} \sum_{q=1}^K \operatorname{diag}(d_q^h) \right) \Phi.$$

$$B_h = \frac{1}{K^2} \sum_{q,q'=1}^K (\Phi^T d_q^h)(\Phi^T d_{q'}^h)^T = \Phi^T \left( \left( \frac{1}{K} \sum_q d_q^h \right) \left( \frac{1}{K} \sum_{q'} d_{q'}^h \right)^T \right) \Phi = \Phi^T (\bar{d}^h)(\bar{d}^h)^T \Phi.$$

Combining $A_h - B_h$ gives the second result. This completes the proof. $\square$

This final expression for $\tilde{I}_h(d_{1:K}^h)$ is independent of $\theta$ and computationally tractable. Therefore, our practical experimental design objective becomes optimizing a scalar criterion $s(\cdot)$ over the state visitation measures $d_{1:K} = \{d_q^h\}_{q \in [K], h \in [H]}$:

$$\underset{d_{1:K}}{\arg\max}\, s\left( T \cdot \sum_{h=1}^{H} \tilde{I}_h(d_{1:K}^h) + \lambda I_d \right) \tag{12}$$

The optimization is subject to the constraints that these visitation measures are valid in the given MDP.

### B.4.1 Information Decomposition and Policy Diversity

The tractable objective derived from Theorem B.3 provides valuable insight into what constitutes an informative experiment in the context of preference-based RL. Let's examine the core matrix term within the approximate FIM at timestep $h$:

$$M_h(d_{1:K}^h) = \frac{1}{K} \sum_{q=1}^{K} \text{diag}(d_q^h) - \bar{d}^h (\bar{d}^h)^T$$

This expression can be interpreted in terms of the statistics of the state visitation distributions. The first term, $\frac{1}{K} \sum_{q=1}^{K} \text{diag}(d_q^h)$, represents the average of the per-policy state variances (since $\text{diag}(d_q^h)$ captures the variance if states were one-hot encoded). The second term, $\bar{d}^h (\bar{d}^h)^T$, represents the outer product of the average state visitation vector. The structure resembles a covariance matrix: $\mathbb{E}[xx^T] - \mathbb{E}[x]\,\mathbb{E}[x]^T$. Maximizing a scalar function of $\tilde{I}_h(d_{1:K}^h) = \Phi^T M_h(d_{1:K}^h)\Phi$ encourages policies whose average state visitation behavior exhibits high spread in feature space, after accounting for the variance of the average distribution.

This suggests that the objective implicitly favors diversity among the chosen policies $\pi_1, \ldots, \pi_K$. If all policies induce very similar state visitation distributions ($d_q^h \approx \bar{d}^h$ for all $q$), the term $M_h(d_{1:K}^h)$ might be small. Conversely, if the policies explore distinct regions of the state space, leading to diverse $d_q^h$ vectors, the resulting $M_h(d_{1:K}^h)$ is likely to be larger (in a matrix sense, e.g., larger eigenvalues), contributing more to the information gain.

This intuition is made precise by Lemma B.4, which provides an alternative decomposition of $\tilde{I}_h(d_{1:K}^h)$. Invoking this lemma, we can rewrite the approximate FIM contribution as:

$$\Phi^T \left[ \underbrace{\frac{1}{K} \sum_{q=1}^{K} \left( \text{diag}(d_q^h) - d_q^h (d_q^h)^T \right)}_{\substack{\text{Average Per-Policy} \\ \text{State Covariance}}} + \underbrace{\frac{1}{K^2} \sum_{1 \le i < j \le K} (d_i^h - d_j^h)(d_i^h - d_j^h)^T}_{\substack{\text{Average Pairwise Difference} \\ \text{(Diversity Term)}}} \right] \Phi$$

This decomposition separates the information contribution into two components:

**Average Per-Policy State Covariance**. The first term represents the average of the covariance matrices associated with each policy's state visitation distribution $d_q^h$. It captures the spread within each policy's behavior at timestep $h$; maximizing it encourages policies that individually explore diverse states within their own trajectories.

**Average Pairwise Difference (Diversity Term)**. The second term directly quantifies diversity between policies. It is the sum of outer products of differences between the visitation vectors of all pairs $(i, j)$. This term is maximized when the distributions $d_i^h$ and $d_j^h$ are distinct, promoting policies that explore complementary regions of the state space.

Therefore, optimizing the approximate FIM objective naturally balances exploring broadly within each policy and ensuring that the set of policies collectively covers different aspects of the state space, maximizing the potential for informative comparisons.

**Lemma B.4.** *Let $\tilde{I}_h(d_{1:K}^h)$ be the approximate expected Fisher Information Matrix contribution at timestep $h$ under the uniform preference assumption, as given in Theorem B.3:*

$$\tilde{I}_h(d_{1:K}^h) = \Phi^T \left( \frac{1}{K} \sum_{q=1}^{K} \text{diag}(d_q^h) - \bar{d}^h (\bar{d}^h)^T \right) \Phi$$

*where $d_q^h \in \mathbb{R}^{|S|}$ is the state visitation vector for policy $\pi_q$ at step $h$, $\Phi \in \mathbb{R}^{|S| \times d}$ is the feature matrix, and $\bar{d}^h = \frac{1}{K} \sum_{q=1}^{K} d_q^h$. This can be rewritten in terms of pairwise differences between state visitation vectors as:*

$$\tilde{I}_h(d_{1:K}^h) = \Phi^T \left[ \frac{1}{K} \sum_{q=1}^{K} \left( \mathrm{diag}(d_q^h) - d_q^h (d_q^h)^T \right) + \frac{1}{K^2} \sum_{1 \leq i < j \leq K} (d_i^h - d_j^h)(d_i^h - d_j^h)^T \right] \Phi$$

*Proof.* We begin with the definition from Theorem B.3. Let $M_h(d_{1:K}^h)$ denote the matrix expression within $\Phi^T(\dots)\Phi$:

$$M_h(d_{1:K}^h) = \frac{1}{K} \sum_{q=1}^{K} \mathrm{diag}(d_q^h) - \bar{d}^h (\bar{d}^h)^T$$

Expand the outer product of the average state visitation vector:

$$\bar{d}^h (\bar{d}^h)^T = \left( \frac{1}{K} \sum_{i=1}^{K} d_i^h \right) \left( \frac{1}{K} \sum_{j=1}^{K} d_j^h \right)^T = \frac{1}{K^2} \sum_{i=1}^{K} \sum_{j=1}^{K} d_i^h (d_j^h)^T$$

Substitute this into the expression for $M_h(d_{1:K}^h)$:

$$M_h(d_{1:K}^h) = \frac{1}{K} \sum_{q=1}^{K} \mathrm{diag}(d_q^h) - \frac{1}{K^2} \sum_{i=1}^{K} \sum_{j=1}^{K} d_i^h (d_j^h)^T$$

We split the double summation based on whether the indices are equal $(i = j)$ or distinct $(i \neq j)$:

$$\sum_{i=1}^{K} \sum_{j=1}^{K} d_i^h (d_j^h)^T = \sum_{q=1}^{K} d_q^h (d_q^h)^T + \sum_{i \neq j} d_i^h (d_j^h)^T$$

Substituting this yields:

$$M_h(d_{1:K}^h) = \frac{1}{K} \sum_{q=1}^{K} \mathrm{diag}(d_q^h) - \frac{1}{K^2} \sum_{q=1}^{K} d_q^h (d_q^h)^T - \frac{1}{K^2} \sum_{i \neq j} d_i^h (d_j^h)^T$$

By adding and subtracting the term $(K-1)\frac{1}{K^2} \sum_{q=1}^{K} d_q^h (d_q^h)^T$:

$$M_h(d_{1:K}^h) = \frac{1}{K} \sum_{q=1}^{K} \mathrm{diag}(d_q^h) - \frac{1}{K^2} \sum_{q=1}^{K} d_q^h (d_q^h)^T - (K-1)\frac{1}{K^2} \sum_{q=1}^{K} d_q^h (d_q^h)^T$$

$$+ (K-1)\frac{1}{K^2} \sum_{q=1}^{K} d_q^h (d_q^h)^T - \frac{1}{K^2} \sum_{i \neq j} d_i^h (d_j^h)^T$$

Combine the terms containing $\sum_{q=1}^{K} d_q^h (d_q^h)^T$:

$$M_h(d_{1:K}^h) = \frac{1}{K} \sum_{q=1}^{K} \text{diag}(d_q^h) - (1 + K - 1)\frac{1}{K^2} \sum_{q=1}^{K} d_q^h (d_q^h)^T$$

$$+ \frac{K-1}{K^2} \sum_{q=1}^{K} d_q^h (d_q^h)^T - \frac{1}{K^2} \sum_{i \neq j} d_i^h (d_j^h)^T$$

$$= \frac{1}{K} \sum_{q=1}^{K} \text{diag}(d_q^h) - \frac{K}{K^2} \sum_{q=1}^{K} d_q^h (d_q^h)^T$$

$$+ \frac{K-1}{K^2} \sum_{q=1}^{K} d_q^h (d_q^h)^T - \frac{1}{K^2} \sum_{i \neq j} d_i^h (d_j^h)^T$$

$$= \left( \frac{1}{K} \sum_{q=1}^{K} \text{diag}(d_q^h) - \frac{1}{K} \sum_{q=1}^{K} d_q^h (d_q^h)^T \right)$$

$$+ \left( \frac{K-1}{K^2} \sum_{q=1}^{K} d_q^h (d_q^h)^T - \frac{1}{K^2} \sum_{i \neq j} d_i^h (d_j^h)^T \right)$$

Consider the sum of outer products of pairwise differences over unique pairs $\{i, j\}$ such that $1 \leq i < j \leq K$:

$$\sum_{1 \leq i < j \leq K} (d_i^h - d_j^h)(d_i^h - d_j^h)^T = \sum_{i<j} \left( d_i^h (d_i^h)^T - d_i^h (d_j^h)^T - d_j^h (d_i^h)^T + d_j^h (d_j^h)^T \right)$$

$$= (K-1) \sum_{q=1}^{K} d_q^h (d_q^h)^T - \sum_{i<j} \left( d_i^h (d_j^h)^T + d_j^h (d_i^h)^T \right)$$

The second term $\sum_{i<j} (d_i^h (d_j^h)^T + d_j^h (d_i^h)^T)$ sums over all distinct pairs $\{i, j\}$, equivalent to the summation $\sum_{i \neq j} d_i^h (d_j^h)^T$. Thus,

$$\sum_{1 \leq i < j \leq K} (d_i^h - d_j^h)(d_i^h - d_j^h)^T = (K-1) \sum_{q=1}^{K} d_q^h (d_q^h)^T - \sum_{i \neq j} d_i^h (d_j^h)^T$$

Dividing by $K^2$ yields:

$$\frac{1}{K^2} \sum_{1 \leq i < j \leq K} (d_i^h - d_j^h)(d_i^h - d_j^h)^T = \frac{K-1}{K^2} \sum_{q=1}^{K} d_q^h (d_q^h)^T - \frac{1}{K^2} \sum_{i \neq j} d_i^h (d_j^h)^T$$

This exactly matches the second grouped term derived for $M_h(d_{1:K}^h)$. Substituting this structure back gives:

$$M_h(d_{1:K}^h) = \left( \frac{1}{K} \sum_{q=1}^{K} \text{diag}(d_q^h) - \frac{1}{K} \sum_{q=1}^{K} d_q^h (d_q^h)^T \right) + \frac{1}{K^2} \sum_{1 \leq i < j \leq K} (d_i^h - d_j^h)(d_i^h - d_j^h)^T$$

$$= \frac{1}{K} \sum_{q=1}^{K} \left( \text{diag}(d_q^h) - d_q^h (d_q^h)^T \right) + \frac{1}{K^2} \sum_{1 \leq i < j \leq K} (d_i^h - d_j^h)(d_i^h - d_j^h)^T$$

Finally, reintroducing the outer feature matrix multiplication provides the desired result:

$$\tilde{I}_h(d_{1:K}^h) = \Phi^T M_h(d_{1:K}^h)\Phi = \Phi^T \left[ \frac{1}{K} \sum_{q=1}^{K} \left( \text{diag}(d_q^h) - d_q^h (d_q^h)^T \right) + \frac{1}{K^2} \sum_{1 \leq i < j \leq K} (d_i^h - d_j^h)(d_i^h - d_j^h)^T \right] \Phi$$

$\square$

## B.5 FORMAL: RELATIONSHIP BETWEEN THE STATE-BASED FEEDBACK MODEL AND THE TRUNCATED FEEDBACK MODEL

We now formally analyze the relationship between the information content of the State-based feedback model and the Truncated Trajectory feedback model. This analysis is performed under the *perfect decomposition condition*, where the features of a truncated trajectory are assumed to be the sum of the features of its constituent states. Additionally, we utilize the uniform preference approximation $(p(q|s_{1..K}) \approx 1/K)$ introduced in Step 2 of Section 4.3 (Eq. 10), which yields the following $\theta$-independent structure for the approximated Fisher Information matrix component, $\tilde{I}_h$, derived from comparing $K$ feature vectors $\{\mathbf{x}_q\}_{q=1}^K$ at a given step $h$:

$$\tilde{I}_h(\mathbf{x}_1, \ldots, \mathbf{x}_K) = \frac{1}{K} \sum_{q=1}^K \mathbf{x}_q \mathbf{x}_q^\top - \frac{1}{K^2} \sum_{q,q'=1}^K \mathbf{x}_q \mathbf{x}_{q'}^\top$$

The following theorem compares the approximated FIMs of the two models under these conditions.

**Theorem B.5** (Comparison of Approximated FIMs under Perfect Decomposition)**.** *Let* $\mathcal{T}^K = \{(\tau_t^1, \ldots, \tau_t^K)\}_{t=1}^T$ *be a set of* $T \times K$ *trajectories. For the standard (state-based) feedback model, let* $\phi(s_{t,h}^q)$ *be the feature vector for the state* $s_{t,h}^q$*. The approximated Fisher Information Matrix is*

$$\tilde{I}^{state}(\mathcal{T}^K) = \sum_{t=1}^T \sum_{h=1}^H \left( \frac{1}{K} \sum_{q=1}^K \phi(s_{t,h}^q)\phi(s_{t,h}^q)^\top - \frac{1}{K^2} \sum_{q,q'=1}^K \phi(s_{t,h}^q)\phi(s_{t,h}^{q'})^\top \right).$$

*For the truncated trajectory feedback model, assume the perfect decomposition condition holds, such that the feature representation of the* $q$*-th trajectory in episode* $t$ *truncated at timestep* $h$ *is* $\psi_{t,h}^q = \sum_{h'=1}^h \phi(s_{t,h'}^q)$*. The corresponding approximated Fisher Information Matrix is*

$$\tilde{I}^{trunc}(\mathcal{T}^K) = \sum_{t=1}^T \sum_{h=1}^H \left( \frac{1}{K} \sum_{q=1}^K \psi_{t,h}^q (\psi_{t,h}^q)^\top - \frac{1}{K^2} \sum_{q,q'=1}^K \psi_{t,h}^q (\psi_{t,h}^{q'})^\top \right).$$

*Then, under the perfect decomposition condition,*

$$\tilde{I}^{trunc}(\mathcal{T}^K) \succeq \frac{1}{4} \cdot \tilde{I}^{state}(\mathcal{T}^K).$$

*Proof of Theorem B.5.* Let $\mathcal{T}^K = \{(\tau_t^1, \ldots, \tau_t^K)\}_{t=1}^T$ be the set of trajectories. We define two approximated Fisher Information Matrices based on the uniform preference assumption $(p \approx 1/K)$.

First, the FIM for the state-based feedback model, denoted $I_{\text{approx}}^{\text{state}}(\mathcal{T}^K)$, uses features $\phi(s_{t,h}^q)$ from individual states:

$$I_{\text{approx}}^{\text{state}}(\mathcal{T}^K) = \sum_{t=1}^T \sum_{h=1}^H \left( \frac{1}{K} \sum_{q=1}^K \phi(s_{t,h}^q)\phi(s_{t,h}^q)^\top - \frac{1}{K^2} \sum_{q,q'=1}^K \phi(s_{t,h}^q)\phi(s_{t,h}^{q'})^\top \right).$$

Next, the FIM for the truncated trajectory feedback model, $I_{\text{approx}}^{\text{trunc}}(\mathcal{T}^K)$, under the perfect decomposition condition, uses the sum-decomposed features $\psi_{t,h}^q = \sum_{h'=1}^h \phi(s_{t,h'}^q)$:

$$I_{\text{approx}}^{\text{trunc}}(\mathcal{T}^K) = \sum_{t=1}^T \sum_{h=1}^H \left( \frac{1}{K} \sum_{q=1}^K \psi_{t,h}^q (\psi_{t,h}^q)^\top - \frac{1}{K^2} \sum_{q,q'=1}^K \psi_{t,h}^q (\psi_{t,h}^{q'})^\top \right).$$

Let $\mathbf{\Phi}_{t,q} \in \mathbb{R}^{H \times d}$ be the matrix whose $h$-th row is $\phi(s_{t,h}^q)^\top$. That is,

$$\mathbf{\Phi}_{t,q} = \begin{pmatrix} \phi(s_{t,1}^q)^\top \\ \phi(s_{t,2}^q)^\top \\ \vdots \\ \phi(s_{t,H}^q)^\top \end{pmatrix}.$$

The sum of outer products over the horizon $H$ for the state-based model is $\sum_{h=1}^{H} \phi(s_{t,h}^q)\phi(s_{t,h}^q)^\top = \mathbf{\Phi}_{t,q}^\top I_H \mathbf{\Phi}_{t,q}$, where $I_H$ is the $H \times H$ identity matrix. Similarly, the sum of cross-products is $\sum_{h=1}^{H} \phi(s_{t,h}^q)\phi(s_{t,h}^{q'})^\top = \mathbf{\Phi}_{t,q}^\top I_H \mathbf{\Phi}_{t,q'}$. Thus, $\tilde{I}^{\text{state}}(\mathcal{T}^K)$ can be rewritten as:

$$\tilde{I}^{\text{state}}(\mathcal{T}^K) = \sum_{t=1}^{T} \left( \frac{1}{K} \sum_{q=1}^{K} \mathbf{\Phi}_{t,q}^\top I_H \mathbf{\Phi}_{t,q} - \frac{1}{K^2} \sum_{q,q'=1}^{K} \mathbf{\Phi}_{t,q}^\top I_H \mathbf{\Phi}_{t,q'} \right).$$

For the truncated trajectory model (under perfect decomposition), let $\mathbf{\Psi}_{t,q} \in \mathbb{R}^{H \times d}$ be the matrix whose $h$-th row is $(\psi_{t,h}^q)^\top = \left( \sum_{h'=1}^{h} \phi(s_{t,h'}^q) \right)^\top$. Let $S \in \mathbb{R}^{H \times H}$ be the lower triangular matrix of ones, i.e., $S_{ij} = 1$ if $i \geq j$ and $S_{ij} = 0$ if $i < j$. For example, if $H = 3$:

$$S = \begin{pmatrix} 1 & 0 & 0 \\ 1 & 1 & 0 \\ 1 & 1 & 1 \end{pmatrix}.$$

The cumulative sum structure means $\mathbf{\Psi}_{t,q} = S\mathbf{\Phi}_{t,q}$. The sum of outer products over the horizon $H$ for the truncated model is $\sum_{h=1}^{H} \psi_{t,h}^q(\psi_{t,h}^q)^\top = \mathbf{\Psi}_{t,q}^\top \mathbf{\Psi}_{t,q} = (S\mathbf{\Phi}_{t,q})^\top(S\mathbf{\Phi}_{t,q}) = \mathbf{\Phi}_{t,q}^\top S^\top S \mathbf{\Phi}_{t,q}$. Similarly, the sum of cross-products is $\sum_{h=1}^{H} \psi_{t,h}^q(\psi_{t,h}^{q'})^\top = \mathbf{\Psi}_{t,q}^\top \mathbf{\Psi}_{t,q'} = \mathbf{\Phi}_{t,q}^\top S^\top S \mathbf{\Phi}_{t,q'}$. Let $M = S^\top S$. This is an $H \times H$ symmetric positive definite matrix. For $H = 3$, $M = S^\top S = \begin{pmatrix} 1 & 1 & 1 \\ 0 & 1 & 1 \\ 0 & 0 & 1 \end{pmatrix} \begin{pmatrix} 1 & 0 & 0 \\ 1 & 1 & 0 \\ 1 & 1 & 1 \end{pmatrix} = \begin{pmatrix} 3 & 2 & 1 \\ 2 & 2 & 1 \\ 1 & 1 & 1 \end{pmatrix}$. Thus, $\tilde{I}^{\text{trunc}}(\mathcal{T}^K)$ can be expressed in terms of $\mathbf{\Phi}_{t,q}$ and $M$:

$$\tilde{I}^{\text{trunc}}(\mathcal{T}^K) = \sum_{t=1}^{T} \left( \frac{1}{K} \sum_{q=1}^{K} \mathbf{\Phi}_{t,q}^\top M \mathbf{\Phi}_{t,q} - \frac{1}{K^2} \sum_{q,q'=1}^{K} \mathbf{\Phi}_{t,q}^\top M \mathbf{\Phi}_{t,q'} \right).$$

Let $X_t = \begin{pmatrix} \mathbf{\Phi}_{t,1}^\top & \cdots & \mathbf{\Phi}_{t,K}^\top \end{pmatrix}^\top \in \mathbb{R}^{(KH) \times d}$. Let $J_K = \frac{1}{K}\mathbf{1}_K\mathbf{1}_K^\top$ be the $K \times K$ matrix of all $1/K$. Let $I_K$ be the $K \times K$ identity matrix. The FIM expressions can be written compactly using Kronecker products $\otimes$:

$$\tilde{I}^{\text{state}}(\mathcal{T}^K) = \sum_{t=1}^{T} X_t^\top \left( \frac{1}{K}(I_K - J_K) \otimes I_H \right) X_t$$

$$\tilde{I}^{\text{trunc}}(\mathcal{T}^K) = \sum_{t=1}^{T} X_t^\top \left( \frac{1}{K}(I_K - J_K) \otimes M \right) X_t$$

Since $M = S^\top S$ is positive definite (as $S$ is invertible), its eigenvalues are positive. Let $\lambda_{min}(M)$ be the smallest eigenvalue of $M$. Then $M \succeq \lambda_{min}(M)I_H$. The matrix $I_K - J_K$ is positive semidefinite (it's proportional to a projection matrix). Therefore, using properties of Kronecker products and Loewner order:

$$(I_K - J_K) \otimes M \succeq (I_K - J_K) \otimes (\lambda_{min}(M)I_H) = \lambda_{min}(M)(I_K - J_K) \otimes I_H$$

Multiplying by $1/K$ and summing over $t$ after pre- and post-multiplying by $X_t^\top$ and $X_t$:

$$\sum_{t=1}^{T} X_t^\top \left( \frac{1}{K}(I_K - J_K) \otimes M \right) X_t \succeq \lambda_{min}(M) \sum_{t=1}^{T} X_t^\top \left( \frac{1}{K}(I_K - J_K) \otimes I_H \right) X_t$$

This shows $\tilde{I}^{\text{trunc}}(\mathcal{T}^K) \succeq \lambda_{min}(M) \cdot \tilde{I}^{\text{state}}(\mathcal{T}^K)$. The eigenvalues of $M = S^\top S$ are known to be $\lambda_k(M) = \frac{1}{4\sin^2\left( \frac{(2k-1)\pi}{2(2H+1)} \right)}$ for $k = 1, \ldots, H$. Since $\sin^2(x) \leq 1$ for any $x$, the minimum eigenvalue $\lambda_{min}(M)$ is lower-bounded by $1/4$. Therefore, we can state the result using this constant lower bound:

$$\tilde{I}^{\text{trunc}}(\mathcal{T}^K) \succeq \frac{1}{4} \cdot \tilde{I}^{\text{state}}(\mathcal{T}^K)$$

This completes the proof. $\qquad\square$

## B.6 DETAILED ALGORITHM DESCRIPTION

Our Experimental Design for Preference-Based Reinforcement Learning (ED-PBRL) algorithm, detailed in Algorithm 2, consists of two main phases. The first phase optimizes a set of $K$ policies using Convex-RL according to our objective derived in Section 4.3. The second phase plays these optimized policies to collect $K$ sets of trajectories for obtaining preferences.

---

**Algorithm 2** ED-PBRL using Convex-RL (Detailed Version of Algorithm 1)

---

**Input:** Known MDP components $M = (\mathcal{S}, \mathcal{A}, P, H, \rho)$, number of policies $K$, number of episodes $T$, feature map $\Phi$, scalar criterion $s(\cdot)$, number of optimization iterations $N$, regularization constant $\lambda$ ($\lambda > 0$)

**Output:** Estimated preference parameter $\hat{\theta}$

**Phase 1: Compute Optimal State Visitation Measures** {Solve Eq. 6}

Initialize $^{(1)}d_{\text{mix},q}^{\{1,\dots,H\}} \leftarrow \mathbf{0}$ for $q = 1, \dots, K$ {Initialize visitation measures}

**for** $n = 1$ **to** $N - 1$ **do**

    Let $I_{total}^{(n)} = T \sum_{h=1}^{H} \tilde{I}_h(^{(n)}d_{\text{mix},1:K}^h) + \lambda I_d$ {Objective using $^{(n)}d_{\text{mix}}$}

    **for** $q = 1$ **to** $K$ **do**

        Compute gradient reward: $r_{\text{grad}_q}(h, s, a) \leftarrow \nabla_{d_{\pi_q}^h(s,a)} s(I_{total}^{(n)})$

        Find policy maximizing linear objective: $\pi_{\text{grad}_q}^{(n)} \leftarrow \texttt{value\_iteration}(M, r_{\text{grad}_q})$

        Compute corresponding visitation vector $d_{\text{grad}_q}^{(n),\{1,\dots,H\}}$ from $\pi_{\text{grad}_q}^{(n)}$

    **end for**

    Determine step size $\alpha_n$ via line search: For $q = 1, \dots, K$, let $d_{\text{cand},q}^h(\alpha') = (1 - \alpha')^{(n)}d_{\text{mix},q}^h + \alpha' d_{\text{grad},q}^{(n),h}$. Find $\alpha_n \leftarrow \arg\max_{\alpha' \in [0,1]} s\left(T \sum_{h=1}^{H} \tilde{I}_h(d_{\text{cand},1:K}^h(\alpha')) + \lambda I_d\right)$ (see Eq. 11 for $\tilde{I}_h$)

    **for** $q = 1$ **to** $K$ **do**

        $^{(n+1)}d_{\text{mix},q}^{\{1,\dots,H\}} \leftarrow (1 - \alpha_n) \cdot {}^{(n)}d_{\text{mix},q}^{\{1,\dots,H\}} + \alpha_n \cdot d_{\text{grad}_q}^{(n),\{1,\dots,H\}}$

    **end for**

**end for**

Let $\{d_{\text{mix},q}^{*h}\}_{h,q} \leftarrow \{^{(N)}d_{\text{mix},q}^{\{1,\dots,H\}}\}_{h,q}$ be the final optimal visitation measures.

**Phase 2: Policy Extraction and Trajectory Sampling**

**for** $q = 1$ **to** $K$ **do**

    Extract policy $\pi_q^*$ from final visitation measure $d_{\text{mix},q}^{*h}$

    $\mathcal{T}_q \leftarrow \emptyset$ {Initialize trajectory set for policy $\pi_q^*$}

**end for**

**for** $t = 1$ **to** $T$ **do**

    **for** $q = 1$ **to** $K$ **do**

        Sample trajectory $\tau_t^q \sim \pi_q^*$

        $\mathcal{T}_q \leftarrow \mathcal{T}_q \cup \{\tau_t^q\}$

    **end for**

**end for**

Let $\mathcal{D}_{feedback} = \{\mathcal{T}_q\}_{q=1}^K$ be the collected trajectories.

**Phase 3: Parameter Estimation**

Collect preference feedback for trajectories in $\mathcal{D}_{feedback}$.

Estimate $\hat{\theta}$ using all collected feedback (cf. Section 3 for estimation equation).

**return** $\hat{\theta}$

---

**Phase 1: Compute Optimal State Visitation Measures** This phase adapts the Frank-Wolfe algorithm Frank & Wolfe (1956) to maximize the objective $s(I_{total}(\pi_{1:K}))$. Here, $I_{total}(\pi_{1:K})$ represents the total approximate expected regularized FIM (the matrix argument of $s(\cdot)$ in Eq. 6), expressed in terms of policy-induced visitation measures. This is achieved by iteratively building state-action visitation measures $\{^{(n)}d_{\text{mix},q}^h\}$ corresponding to conceptual mixture policies. The process starts with $^{(1)}d_{\text{mix},q} = \mathbf{0}$.

Each iteration $n$ of this phase involves these main steps:

1. **Gradient Computation**: The gradient of $s(I_{total})$ (using the current $^{(n)}d_{\text{mix},q}$) defines a reward function $r_{\text{grad}_q}$ for each policy $q$.

2. **Policy Search Oracle**: For each $q$, a new base policy $\pi^{(n)}_{\text{grad}_q}$ is found by maximizing the expected cumulative reward $r_{\text{grad}_q}$ (e.g., via value iteration). Its visitation measure $d^{(n)}_{\text{grad}_q}$ is computed.

3. **Line Search for Step Size**: The optimal step size $\alpha_n$ is determined by maximizing $s(\cdot)$ for the candidate mixture $(1 - \alpha_n)^{(n)}d_{\text{mix},q} + \alpha_n d^{(n)}_{\text{grad}_q}$.

4. **Mixture Update**: The next mixture's visitation measure is constructed: $^{(n+1)}d_{\text{mix},q} \leftarrow (1 - \alpha_n)^{(n)}d_{\text{mix},q} + \alpha_n d^{(n)}_{\text{grad}_q}$. This efficiently computes the visitation measure of the new conceptual mixture policy $\pi^{(n)}_{\text{mix},q}$.

This iterative process converges to the globally optimal visitation measures $\{d^{*h}_{\text{mix},q}\}$ due to the concavity of $s(\cdot)$ and the convexity of the feasible set of visitation measures.

**Phase 2: Policy Extraction and Trajectory Sampling**  Upon convergence of Phase 1 after $N - 1$ iterations, the final policies $\{\pi^*_q\}^K_{q=1}$ are extracted from the resulting state-action visitation measures $\{d^{*h}_{\text{mix},q}\}^K_{q=1}$. These policies are then executed to generate the $K \times T$ trajectories, which form the dataset $\mathcal{D}_{feedback}$ for collecting user preference feedback.

**Phase 3: Parameter Estimation**  After the trajectories are generated and collected into $\mathcal{D}_{feedback}$ in Phase 2, preference feedback is obtained from the user for these trajectories. This accumulated feedback is then used to compute the final estimate $\hat{\theta}$ of the true reward parameter $\theta$, as detailed in Section 3.

### B.7  Detailed Theoretical Guarantees

The Convex-RL optimization phase (Algorithm 2, lines 13-24) employs the Frank-Wolfe algorithm (also known as the conditional gradient method) over the convex polytope of valid state-action visitation measures Puterman (2014); Frank & Wolfe (1956); Jaggi (2013). The inclusion of an exact line search for the step size $\alpha_n$ is a standard variant of the Frank-Wolfe algorithm.

The key to guaranteeing global optimality for this procedure is the concavity of the objective function. Let $D = \{d^h_q\}_{h \in [H], q \in [K]}$ represent the collection of all state visitation vectors, where each $d^h_q \in \Delta^{|\mathcal{S}|-1}$ (the probability simplex over states). The domain of $D$ is a convex set. The objective function is $f(D) = s(I_{total}(D))$, where $I_{total}(D)$ is precisely the matrix argument of $s(\cdot)$ in Eq. 6:

$$I_{total}(D) = T \sum_{h=1}^{H} \left[ \Phi^T \left( \frac{1}{K} \sum_{q=1}^{K} \text{diag}(d^h_q) - \left( \frac{1}{K} \sum_{q=1}^{K} d^h_q \right) \left( \frac{1}{K} \sum_{q=1}^{K} d^h_q \right)^T \right) \Phi \right] + \lambda I_d.$$

With the concavity of the objective function established (Theorem 5.1), we can state the convergence guarantee for Algorithm 2 (Theorem B.6), which implements the Frank-Wolfe method.

#### B.7.1  Proof of Objective Function Concavity (Theorem 5.1)

**Theorem 5.1.** *Let $d_{1:K} = \{d^h_q\}_{h \in [H], q \in [K]}$. If the scalarization $s : \mathbb{S}^d_+ \to \mathbb{R}$ is concave and matrix-monotone, then $\left( I_{total}(d_{1:K}) \right)$ as defined in (6) is concave in $d_{1:K}$.*

*Proof.* Let $D = \{d^h_q\}_{h \in [H], q \in [K]}$ be the collection of state visitation vectors, where each $d^h_q \in \Delta^{|\mathcal{S}|-1}$ (the probability simplex in $\mathbb{R}^{|\mathcal{S}|}$). The domain of $D$, denoted $\mathcal{D}_{sv}$, is a Cartesian product of

simplices, which is a convex set. The objective function is $f(D) = s(I_{total}(D))$, where

$$I_{total}(D) = T \sum_{h=1}^{H} \tilde{I}_h(D^h) + \lambda I_d,$$

which coincides with the definition inside Eq. 6 in the main text. and $D^h = (d_1^h, \ldots, d_K^h)$ are the visitation vectors for timestep $h$. The term $\tilde{I}_h(D^h)$ is given by:

$$\tilde{I}_h(D^h) = \Phi^T M_h(D^h)\Phi, \quad \text{with} \quad M_h(D^h) = \frac{1}{K} \sum_{q=1}^{K} \text{diag}(d_q^h) - \left(\frac{1}{K}\sum_{q=1}^{K} d_q^h\right)\left(\frac{1}{K}\sum_{q=1}^{K} d_q^h\right)^T.$$

We will prove the concavity of $f(D)$ by showing that $I_{total}(D)$ is a concave matrix-valued function of $D$, and then using the properties of $s(\cdot)$.

1. **Concavity of** $M_h(D^h)$: Let $L_h(D^h) = \frac{1}{K}\sum_{q=1}^{K}\text{diag}(d_q^h)$. The function $\text{diag}(v)$ is a linear mapping from a vector $v$ to a diagonal matrix. Thus, $L_h(D^h)$ is a linear function of the collection of vectors $D^h = (d_1^h, \ldots, d_K^h)$. Linear functions are both concave and convex.

Let $\bar{d}^h(D^h) = \frac{1}{K}\sum_{q=1}^{K} d_q^h$. This is also a linear function of $D^h$. Consider the function $Q(v) = -vv^T$. The function $v \mapsto vv^T$ is convex. To see this, for $v_1, v_2$ and $\alpha \in [0,1]$:

$$\alpha v_1 v_1^T + (1-\alpha)v_2 v_2^T - (\alpha v_1 + (1-\alpha)v_2)(\alpha v_1 + (1-\alpha)v_2)^T$$
$$= \alpha v_1 v_1^T + (1-\alpha)v_2 v_2^T - (\alpha^2 v_1 v_1^T + \alpha(1-\alpha)(v_1 v_2^T + v_2 v_1^T) + (1-\alpha)^2 v_2 v_2^T)$$
$$= (\alpha - \alpha^2)v_1 v_1^T + ((1-\alpha) - (1-\alpha)^2)v_2 v_2^T - \alpha(1-\alpha)(v_1 v_2^T + v_2 v_1^T)$$
$$= \alpha(1-\alpha)v_1 v_1^T + \alpha(1-\alpha)v_2 v_2^T - \alpha(1-\alpha)(v_1 v_2^T + v_2 v_1^T)$$
$$= \alpha(1-\alpha)(v_1 - v_2)(v_1 - v_2)^T.$$

Since $\alpha(1-\alpha) \geq 0$ and $(v_1 - v_2)(v_1 - v_2)^T \succeq 0$ (it's an outer product, hence positive semidefinite), the expression is $\succeq 0$. Thus, $v \mapsto vv^T$ is convex. Therefore, $Q(v) = -vv^T$ is concave. The composition of a concave function with a linear function is concave. Since $Q(v)$ is concave and $\bar{d}^h(D^h)$ is linear, the function $D^h \mapsto Q(\bar{d}^h(D^h)) = -\bar{d}^h(D^h)(\bar{d}^h(D^h))^T$ is concave.

$M_h(D^h) = L_h(D^h) + Q(\bar{d}^h(D^h))$ is the sum of a linear function (which is concave) and a concave function. Thus, $M_h(D^h)$ is a concave matrix-valued function of $D^h$.

2. **Concavity of** $\tilde{I}_h(D^h)$: The function $\tilde{I}_h(D^h) = \Phi^T M_h(D^h)\Phi$ is a congruence transformation of $M_h(D^h)$. Since $M_h(D^h)$ is concave in $D^h$, and congruence transformations preserve concavity (i.e., if $A(x)$ is concave, then $C^T A(x)C$ is concave for any constant matrix $C$), $\tilde{I}_h(D^h)$ is concave in $D^h$.

3. **Concavity of** $I_{total}(D)$: The total approximate FIM before regularization is $\sum_{h=1}^{H}\tilde{I}_h(D^h)$. Since each $\tilde{I}_h(D^h)$ is concave with respect to its arguments $D^h$ (and thus with respect to the full $D$, as it doesn't depend on $D^{h'}$ for $h' \neq h$), their sum is concave with respect to $D$. Multiplying by a non-negative scalar $T$ preserves concavity. So, $T\sum_{h=1}^{H}\tilde{I}_h(D^h)$ is concave in $D$. Adding a constant matrix $\lambda I_d$ also preserves concavity. Therefore, $I_{total}(D) = T\sum_{h=1}^{H}\tilde{I}_h(D^h) + \lambda I_d$ is a concave matrix-valued function of $D$.

4. **Concavity of** $s(I_{total}(D))$: We are given that the scalar criterion $s : \mathbb{S}_+^d \to \mathbb{R}$ is concave and matrix-monotone non-decreasing. If $g(x)$ is a matrix-valued concave function and $s(A)$ is a scalar-valued concave and non-decreasing function of matrix $A$ (in the Loewner order), then the composition $s(g(x))$ is concave (see Boyd & Vandenberghe, Convex Optimization, Section 3.2.4). In our case, $g(D) = I_{total}(D)$ is concave in $D$. Thus, $f(D) = s(I_{total}(D))$ is concave with respect to $D = \{d_q^h\}_{h\in[H], q\in[K]}$ over the convex domain $\mathcal{D}_{sv}$. $\qquad \square$

### B.7.2 PROOF OF CONVERGENCE GUARANTEE (THEOREM B.6)

**Theorem B.6.** *[Convergence Guarantee of Algorithm 2 (Detailed)] Let $D^{(n)}$ be the sequence of collections of state visitation measures generated by Algorithm 2, where $D^{(1)}$ is the initialization and*

$D^{(n+1)}$ is the iterate after $n$ Frank-Wolfe steps. Let $f(D) = s(I_{total}(D))$ be the objective function defined in Theorem 5.1, and let $D^* \in \mathcal{D}_{sv}$ be an optimal solution, $D^* = \arg\max_{D \in \mathcal{D}_{sv}} f(D)$. The domain $\mathcal{D}_{sv}$ of valid collections of state visitation measures is compact and convex. If Algorithm 2 performs $N_{iter}$ iterations of the Frank-Wolfe update (i.e., the loop from $n = 1$ to $N_{iter}$ in the algorithm's notation, resulting in the final iterate $D^{(N_{iter}+1)}$), using exact line search for $\alpha_n$ at each iteration, then the suboptimality of the final iterate $D^{(N_{iter}+1)}$ is bounded by:

$$f(D^*) - f(D^{(N_{iter}+1)}) \leq \frac{2C_f}{N_{iter} + 2}$$

where $C_f$ is the curvature constant of $f$ over $\mathcal{D}_{sv}$.

**Theorem B.6.** *[Convergence Guarantee of Algorithm 2 (Detailed)] Let $D^{(n)}$ be the sequence of collections of state visitation measures generated by Algorithm 2, where $D^{(1)}$ is the initialization and $D^{(n+1)}$ is the iterate after $n$ Frank-Wolfe steps. Let $f(D) = s(I_{total}(D))$ be the objective function defined in Theorem 5.1, and let $D^* \in \mathcal{D}_{sv}$ be an optimal solution, $D^* = \arg\max_{D \in \mathcal{D}_{sv}} f(D)$. The domain $\mathcal{D}_{sv}$ of valid collections of state visitation measures is compact and convex. If Algorithm 2 performs $N_{iter}$ iterations of the Frank-Wolfe update (i.e., the loop from $n = 1$ to $N_{iter}$ in the algorithm's notation, resulting in the final iterate $D^{(N_{iter}+1)}$), using exact line search for $\alpha_n$ at each iteration, then the suboptimality of the final iterate $D^{(N_{iter}+1)}$ is bounded by:*

$$f(D^*) - f(D^{(N_{iter}+1)}) \leq \frac{2C_f}{N_{iter} + 2}$$

*where $C_f$ is the curvature constant of $f$ over $\mathcal{D}_{sv}$.*

*Proof.* The convergence of Algorithm 2 relies on standard results for the Frank-Wolfe algorithm when maximizing a concave function over a compact convex set. We verify the conditions required for these guarantees.

**1. Objective Function and Domain:**

- **Concavity:** The objective function $f(D) = s(I_{total}(D))$ is concave with respect to the collection of state visitation vectors $D = \{d_q^h\}_{h,q}$, as proven in Theorem 5.1.

- **Domain $\mathcal{D}_{sv}$:** The domain $\mathcal{D}_{sv}$ is the set of all valid collections of state visitation measures $\{d_q^h\}_{h,q}$. Each $d_q^h$ is a probability distribution over the finite state space $\mathcal{S}$, so it belongs to the probability simplex $\Delta^{|\mathcal{S}|-1}$. The full domain $\mathcal{D}_{sv}$ is a Cartesian product of $K \times H$ such simplices. Each simplex is compact and convex, and thus their Cartesian product $\mathcal{D}_{sv}$ is also compact and convex.

**2. Frank-Wolfe Algorithm Steps:** Algorithm 2 implements the Frank-Wolfe algorithm:

- **Initialization (Line 13):** $^{(1)}d_{\text{mix},q} \leftarrow \mathbf{0}$. This initializes the iterate $D^{(1)}$ within $\mathcal{D}_{sv}$ (the zero vector is on the boundary of the simplex if non-negativity is the only constraint, or can be seen as a valid (degenerate) visitation measure).

- **Gradient Computation (Line 16):** The algorithm computes the gradient $\nabla f(D^{(n)})$ (implicitly, by computing $r_{\text{grad}_q}$ which is derived from this gradient).

- **Linear Maximization Oracle (LMO) (Lines 17-18):** For each $q$, the step $\pi_{\text{grad}_q}^{(n)} \leftarrow$ `value_iteration`$(M, r_{\text{grad}_q})$ finds a policy that maximizes the linear objective $\sum_{s,a} d_\pi^h(s,a) r_{\text{grad}_q}(h, s, a)$ over all policies $\pi$. This is equivalent to finding a vertex $S_q^{(n)}$ of the polytope of visitation measures for policy $q$ that maximizes $\langle \nabla_{d_q^h} f(D^{(n)}), S_q^{(n)} \rangle$. The collection of these $S_q^{(n)}$ for all $q$ forms the $S^{(n)}$ in the standard Frank-Wolfe update $S^{(n)} = \arg\max_{S \in \mathcal{D}_{sv}} \langle \nabla f(D^{(n)}), S \rangle$. The computation of $d_{\text{grad}_q}^{(n)}$ from $\pi_{\text{grad}_q}^{(n)}$ yields this $S^{(n)}$.

- **Step Size (Line 20):** $\alpha_n$ is determined by exact line search: $\alpha_n \leftarrow \arg\max_{\alpha' \in [0,1]} f((1 - \alpha')D^{(n)} + \alpha' S^{(n)})$.

- **Update (Line 22):** $D^{(n+1)} \leftarrow (1 - \alpha_n)D^{(n)} + \alpha_n S^{(n)}$.

The algorithm performs $N_{iter} = N - 1$ such iterations, producing iterates $D^{(2)}, \ldots, D^{(N)}$. (Note: $D^{(1)}$ is the initialization).

**3. Convergence Rate:** For a concave function $f$ maximized over a compact convex set $\mathcal{D}$ using the Frank-Wolfe algorithm with exact line search for the step size, the suboptimality gap $h_k = f(D^*) - f(D^{(k+1)})$ after $k$ iterations (where $D^{(1)}$ is the initial point and $D^{(k+1)}$ is the iterate after $k$ Frank-Wolfe steps) is bounded by (Jaggi, 2013, Theorem 1 and discussion for line search):

$$f(D^*) - f(D^{(k+1)}) \leq \frac{2C_f}{k+2}$$

where $C_f$ is the curvature constant of $f$ over $\mathcal{D}$, defined as

$$C_f = \sup_{\substack{X, S \in \mathcal{D}, \gamma \in (0,1] \\ Y = (1-\gamma)X + \gamma S}} \frac{2}{\gamma^2}(f(X) + \gamma\langle \nabla f(X), S - X \rangle - f(Y)).$$

In our case, Algorithm 2 initializes with $D^{(1)}$ and performs $N_{iter}$ iterations of the Frank-Wolfe update (corresponding to the loop variable $n$ from 1 to $N_{iter}$ in the algorithm's notation as per Algorithm 2 where the loop runs $N - 1$ times; here we use $N_{iter}$ to denote this count of iterations). The final iterate is $D^{(N_{iter}+1)}$. The standard bound $2C_f/(k+2)$ applies after $k$ iterations. Here, $k = N_{iter}$. So, the suboptimality of the final iterate $D^{(N_{iter}+1)}$ is bounded by:

$$f(D^*) - f(D^{(N_{iter}+1)}) \leq \frac{2C_f}{N_{iter} + 2}$$

This holds for $N_{iter} \geq 1$. The constant $C_f$ depends on the objective function $f$ and the domain $\mathcal{D}_{sv}$. Since $\mathcal{D}_{sv}$ is compact, $C_f$ is well-defined and finite, provided $f$ is continuously differentiable (which it is, assuming $s(\cdot)$ is, and $I_{total}(D)$ is differentiable).

Thus, Algorithm 2 converges to the global optimum with a rate of $O(1/N_{iter})$. $\square$

