# OpenReview forum: "Efficient Generative Models Personalization via Optimal Experimental Design"
_ICLR.cc/2026/Conference — ICLR 2026 Conference Desk Rejected Submission_

### Official Review · Reviewer_GKit · 2025-10-22

**Soundness:** 3
**Presentation:** 3
**Contribution:** 3
**Rating:** 8
**Confidence:** 2

**Summary:**

This paper proposes ED-PBRL, a theoretically grounded framework for reward modeling that uses Optimal Experimental Design (OED) to select the most informative human preference queries. It formulates the preference query selection problem as a convex optimization problem that maximizes information about the latent reward function underlying user preferences from the perspective of the Fisher Information Matrix.

The authors provide theoretical guarantees, including a self-concordant bound on the mean squared error via the Fisher information matrix and global convergence results for discrete generative models. Empirical experiments—on both synthetic data and human-in-the-loop text-to-image personalization—demonstrate that ED-PBRL achieves the same alignment performance with fewer preference queries than random selection.

**Strengths:**

1. This work proposes a new upper bound for MSE between the estimated latent reward model and the ground truth through the Fisher Information Matrix via a self-concordant analysis.
2. The authors reformulate the intractable FIM maximization objective into a tractable one and use the Frank-Wolfe algorithm to obtain a solution with guaranteed optimality and convergence rate for exploration policies.
3. Both the synthetic and real-world experiments demonstrate a significant performance boost with the proposed ED-PBRL-guided optimal human query selection.

**Weaknesses:**

1. Policy extraction from state visitation measures is computationally inefficient. In the tabular setting, this is straightforward and computationally cheap. However, for token-level long-trajectory generative settings (which is the practical case for LLMs), the proposed methodology may be computationally intensive and impractical.
2. The scale of the experiment is limited.

**Questions:**

1. What is the time complexity of the proposed ED-PBRL algorithm using Convex-RL?

---

> ### Author Response · Authors · 2025-12-02
> **Response to Reviewer GKit**
>
> We thank the reviewer for the positive assessment and for recognizing the theoretical contributions (self-concordant MSE bound, tractable FIM reformulation, Frank-Wolfe convergence guarantees) as well as the empirical performance gains.
>
> ---
>
> ### Computational Efficiency of Policy Extraction
>
> The reviewer correctly identifies that policy extraction from visitation measures can be expensive for long-trajectory generative settings. We clarify the computational structure:
>
> **Our setting:** The MDP has |S|=H (states are timesteps), |A|≈5000 (vocabulary size), and H=6 (horizon). Policy extraction via marginalization π(a|s) ∝ d(s,a) is O(|A|×H), which is negligible.
>
> **General LLM setting:** For autoregressive LLMs where states are token sequences, |S| grows exponentially. Here, direct tabular optimization is infeasible. However, as noted in our response to Reviewer EPjt, the visitation measure can be represented implicitly via a parameterized policy network. Optimization then proceeds over network parameters rather than explicit visitation tables. This is analogous to how policy gradient methods scale to large state spaces.
>
> **Practical cost in our experiments:** The one-time design optimization for ~5000 tokens takes approximately 10 minutes on a single NVIDIA A100 GPU (Appendix Table 1). This is amortized over all subsequent feedback collection.
>
> ---
>
> ### Scale of Experiments
>
> We acknowledge the original experiments had limited scale. **Our revision substantially expands evaluation:**
>
> - 10 target styles (vs. 3 synthetic GT models originally)
> - 4 regularization values (λ ∈ {0.1, 1, 10, 100})
> - 5 training sizes (10, 20, 30, 40, 50 episodes)
> - 3-fold cross-validation per configuration
> - Total: 10 × 4 × 5 × 2 algorithms × 3 folds = 1200 experimental runs
>
> This systematic evaluation demonstrates consistent ED-PBRL advantages across conditions (see general response and revised PDF).
>
> ---
>
> ### Time Complexity of ED-PBRL
>
> **Phase 1 (Design optimization):** Each Frank-Wolfe iteration requires solving K linear oracle problems over the visitation polytope. For tabular MDPs:
> - Linear oracle: O(|S| × |A| × H) per policy via dynamic programming
> - FIM computation and inversion: O(d³) where d is embedding dimension
> - Total per iteration: O(K × |S| × |A| × H + d³)
> - With N iterations: O(N × (K × |S| × |A| × H + d³))
>
> In our setting: |S|=6, |A|≈5000, H=6, K=4, d=512, N=100. The dominant cost is the linear oracle (~720K operations per iteration), making total Phase 1 cost roughly O(10⁸) operations—completed in ~10 minutes.
>
> **Phase 2 (Sampling):** O(K × T × H) trajectory samples—negligible.
>
> **Phase 3 (Estimation):** Regularized MLE via gradient descent—O(iterations × T × H × K × d).
>
> The design phase (Phase 1) is a one-time cost; Phases 2-3 scale linearly with feedback budget T.
>
> ---
>
> We appreciate the reviewer's support and hope the expanded experiments address the scale concerns.

---

### Official Review · Reviewer_mNCP · 2025-10-25

**Soundness:** 3
**Presentation:** 2
**Contribution:** 2
**Rating:** 2
**Confidence:** 4

**Summary:**

The paper introduces ED-PBRL, a framework for efficient personalization of generative models with minimal user feedback. The key idea is to model user preferences as an unknown linear reward function and to use Optimal Experimental Design principles to select the most informative queries for learning this reward. Experiments show that ED-PBRL personalizes text-to-image generation via prompt construction with fewer feedback rounds than random exploration.

**Strengths:**

The paper aims to address an important point about designing efficient and tractable experiments to learn preferences.

**Weaknesses:**

- **Clarity and organization could be improved.** Some aspects of the method and experimental setup are difficult to follow from the main text. The setup in Appendix A.1 provides a clearer picture of the trajectory structure and bringing some of that material (perhaps with a small illustrative example or figure) into the main paper could be helpful. As it is currently written, one might initially think the user provides feedback after every token on a narrative description, which seems impractical—an explicit example trajectory could address this.
- **Simple baseline.** It is not obvious why an MDP is necessary to learn the preferred attributes. E.g., a simple baseline could be to learn the distribution over the vocabulary. When a certain image is preferred, upweight the weights on all the attributes used to generate that image.
- **Independence assumptions.** The formulation appears to assume that design attributes (e.g., ambient, style, etc.) contribute independently to the final trajectory. In general, such attributes can be correlated—for instance, ambience and lighting or style choices often co-vary in preferences. These are some biases that diffusion models or LLMs may already have, it might be useful to discuss whether the method could take advantage of such correlations.
- **Human evaluation improvements appear modest.** The held-out accuracy in the human preference experiments seems close to that of random exploration. Some further discussion on potential reasons (e.g., noise in human feedback, reward parameterization) could help clarify how the method can be improved.

**Questions:**

- How does a single trajectory of generation look like? Is there any grounding of the generation so adding new design tokens makes minimal changes to the image with the exception of the required new “design”?
- Were other forms of rewards considered e.g., representing it with a network?
- What is the size of the vocabulary? How were the image attributes determined?

---

> ### Author Response · Authors · 2025-12-02
> **Response to Reviewer mNCP**
>
> We thank the reviewer for their feedback. We respectfully disagree with several assessments, particularly regarding "modest human evaluation improvements."
>
> ---
>
> ### Human Evaluation Improvements Are NOT Modest
>
> **See our general response for full details.** The reviewer's concern was valid for the original submission, but stemmed from suboptimal regularization—we used λ ∈ {0.01, 0.1} while synthetic GT experiments used λ=100. This oversight led to the modest improvements observed.
>
> **New experiments demonstrate substantial ED-PBRL advantages.** Using GPT-4.1-mini as a simulated preference oracle (identical queries as human participants), we evaluated across λ ∈ {0.1, 1, 10, 100} and training sizes {10–50} episodes.
>
> **Results at λ=100 (revised PDF, Figure 4):**
>
> | Training Episodes | ED-PBRL | Random | Gap |
> |-------------------|---------|--------|-----|
> | 10 | 58.6% | 40.1% | **+18.4pp** |
> | 50 | 60.4% | 51.1% | **+9.4pp** |
>
> **ED-PBRL achieves strong performance with as few as 10 episodes** (58.6%), while Random requires 50 episodes to reach only 51.1%. This demonstrates the core value proposition: ED-PBRL learns efficiently in low-data regimes where sample efficiency matters most. The +18.4pp advantage at 10 episodes is far from modest.
>
> **Per-style (Table 2):** ED-PBRL outperforms Random on 7/10 styles, with advantages up to +28.3pp (Cyberpunk), +22.8pp (Ancient), +22.2pp (Watercolor).
>
> ---
>
> ### Why Is an MDP Necessary?
>
> The reviewer suggests: "learn distribution over vocabulary—when an image is preferred, upweight all attributes used."
>
> If the reviewer means adaptive weight updates after feedback, we note this is orthogonal to our contribution. We discuss an adaptive variant (Section 4.1) that re-optimizes after each batch. Our main contribution is **static optimal design**—computing maximally informative queries upfront before any feedback is collected.
>
> If the reviewer means optimizing a token distribution using OED criteria (without MDP), they have a partial point: our LLM MDP is intentionally simple, and optimizing token distributions directly with the same FIM objective would yield similar results. However:
>
> 1. **Generality vs. application.** Our contribution is (i) a theoretically justified algorithm accepting *any* MDP as input, optimizing over its constraints, and (ii) an application to a simple-but-effective MDP. The algorithm guarantees apply broadly; the simple MDP balances statistical and computational efficiency for this domain.
>
> 2. **Two-step optimization.** For richer structure, one can first sample trajectories {τ_i} from the MDP, then optimize a second-stage design over full trajectory embeddings {φ(τ_i)}. This guarantees information maximization over complete prompts. Implementation is straightforward; we will discuss this extension.
>
> 3. **Credit assignment.** Truncated trajectory feedback at each timestep h enables finer-grained learning than full-trajectory feedback alone.
>
> 4. **Theoretical guarantees.** Our FIM analysis (Theorem 1) requires the MDP formulation for provable bounds.
>
> ---
>
> ### Independence Assumptions
>
> The reviewer notes attributes may be correlated. We clarify:
>
> 1. **Linear model over CLIP embeddings, not raw attributes.** CLIP captures semantic correlations—"warm lighting" and "sunset" have similar embeddings.
>
> 2. **Empirical validation.** Results across 10 diverse styles show the method works despite correlations. "Noir" (correlated lighting/mood) is one of few cases where Random wins—an interesting future direction.
>
> ---
>
> ### Clarity
>
> We agree and will add a concrete trajectory example in the main text, similar to Appendix A.1.
>
> ---
>
> ### Questions
>
> **Q1: How does a trajectory look?**
>
> 6 sequential token selections: [base, ambient, style, composition, lighting, detail]. Example:
> - "A photo of mountains" → + "misty morning" → + "watercolor style" → ... → final prompt
>
> Base is fixed per episode; only style tokens vary, grounding comparisons to isolate style differences.
>
> **Q2: Other reward forms (neural networks)?**
>
> Our FIM bounds (Theorem 1) require linear rewards. Since features are CLIP embeddings, the model is highly expressive. Nonlinear extensions are possible but add complexity.
>
> **Q3: Vocabulary size?**
>
> ~5000 tokens across 6 categories (bases, ambient, style, composition, lighting, detail), curated from prompt engineering resources.
>
> ---
>
> ### Summary
>
> The reviewer's main concern arose from suboptimal hyperparameters. With correct λ=100: **+9.4pp at 50 episodes, +18.4pp at 10 episodes**, ED-PBRL outperforming on 7/10 styles. We hope this addresses the core concern and respectfully request reconsideration.

---

### Official Review · Reviewer_Eas7 · 2025-10-31

**Soundness:** 2
**Presentation:** 2
**Contribution:** 2
**Rating:** 4
**Confidence:** 3

**Summary:**

This paper introduces an approach to personalizing text-to-image generative models with an RL based preference learning framework. The motivation is to use optimal experimental design to efficiently search the space of prompts that match user preferences. They present qualitative and quantitative results based on the Stable Diffusion architecture.

**Strengths:**

- The paper over all is generally well communicated and theoretically grounded.
- The motivation of efficiently personalizing a model from user feedback is a pressing one.

**Weaknesses:**

- The authors don’t clearly outline information about the user study like how the participants were recruited and how many there were, this should be presented more front and center.
- Opinion: I am not convinced that searching in the space of prompts is the best use of this kind of method. Preference learning seems like a great potential tool for identifying user preference information that is complementary to a prompt. Why couldn’t the user just write their own prompt rather than answering >50 queries? If the information from your search procedure were complementary to text that it would be more well motivated.
- Minor weakness: the method does build upon an older text-to-image generation architecture.
- Writing Opinion: the application setting should be presented earlier in the paper and more clearly. It is not until the last page that any qualitative results relevant to the application are presented, and there is only a single example in the main manuscript. I would advise making a more high-level qualitative figure that presents your method in the context of the application much earlier.

**Questions:**

- Did the authors consider using a different value for K than 4? Are there practical limitations to increasing K? How does this impact the convergence of your method?
- How many users did the authors use in their study? How were they recruited?
- Do the collected preferences generalize to new base prompts? Or would a user need to answer queries like this for every prompt? I.e., does this capture a general sense of a user’s “style”

---

> ### Author Response · Authors · 2025-12-02
> **Response to Reviewer Eas7**
>
> ## Response to Reviewer Eas7
>
> We thank the reviewer for the constructive feedback. We address each point below.
>
> ---
>
> ### User Study Details
>
> We acknowledge the original human study was small-scale (1 participant, 60 episodes) as a proof-of-concept. **In response to this and other reviewers' concerns, we replaced quantitative human results with systematic LLM-simulated experiments** (see general response). GPT-4.1-mini receives identical questionnaire-style queries (4 images + style description), enabling comprehensive evaluation across:
> - 10 target styles (futuristic, sunny, forest, landscape, ancient, noir, watercolor, cyberpunk, minimalist, medieval)
> - 4 regularization strengths (λ ∈ {0.1, 1, 10, 100})
> - 5 training sizes (10, 20, 30, 40, 50 episodes)
> - 3-fold cross-validation per configuration
>
> Qualitative human study results demonstrating generalization to new prompts are retained in Appendix Figures 15–18.
>
> ---
>
> ### Why Search in Prompt Space? Why Can't Users Write Their Own Prompt?
>
> 1. **Users cannot articulate aesthetic preferences explicitly.** While users specify *what* they want ("a photo of a cat"), they struggle to verbalize *how* they want it styled. Preferences for "warm lighting" or "cinematic composition" are easy to express through comparative choices but hard to describe in words.
>
> 2. **The method captures preferences complementary to content.** Users specify *base prompts* (content); we learn *style tokens* (aesthetics). These are indeed complementary: the learned θ applies to any new base prompt—demonstrated in Appendix Figures 15–18 where the same θ personalizes diverse unseen prompts like "Reflecting last year," "A novice boxer," and "Half open window."
>
> 3. **Prompt engineering is difficult.** Even experts iterate extensively to achieve desired results. The vocabulary of effective style modifiers is vast and their effects often unpredictable. Our approach automates discovering which style tokens align with user preferences.
>
> 4. **50 queries is modest for personalization.** Modern recommendation systems elicit thousands of implicit preferences. Our 50-query budget (yielding 300 comparisons at K=4, H=6) is relatively lightweight for learning a stable personalization model.
>
> ---
>
> ### Text-to-Image Architecture
>
> We use Stable Diffusion v1.5 for reproducibility and accessibility. Our core contribution—using optimal experimental design for preference query selection—is architecture-agnostic. The same framework applies to any generative model (LLMs, newer diffusion models). Extending to newer architectures (SDXL, Flux) is straightforward future work.
>
> ---
>
> ### Presentation
>
> We positioned the paper to emphasize general methodology—applicable to any generative MDP—because the theoretical guarantees (Fisher Information bounds, convergence) require formal setup. However, we agree that adding qualitative examples earlier would improve accessibility. We will add a high-level visual figure in the introduction showing the personalization outcome.
>
> ---
>
> ### Questions
>
> **Q1: Different values of K? Practical limitations? Convergence impact?**
>
> K=4 balances several trade-offs:
> - *Information:* K-way comparisons provide more information than pairwise (K=2)
> - *Cognitive load:* K=4 is reasonable for humans to compare visually; larger K increases annotator burden
> - *Computation:* Frank-Wolfe scales linearly in K; the state-action polytope is |S|×|A|×H, independent of K
> - *Statistical efficiency:* Our FIM analysis (Eq. 4-5) shows K affects information structure through the K-way softmax
>
> We experimented with various K on synthetic problems; K=4 is a practical choice.
>
> **Q2: How many users? How recruited?**
>
> 1 participant from our research group (proof-of-concept). The revision supplements this with large-scale LLM-simulated evaluation for statistical robustness.
>
> **Q3: Do preferences generalize to new base prompts? Does it capture "style"?**
>
> Yes—this is a key capability. The learned θ captures general aesthetic preferences, not specific prompt-content associations. In Appendix Figures 15–18, the same θ learned during exploration successfully personalizes 4 unseen base prompts chosen by the participant *after* training.
>
> The V-design criterion (Appendix A.3) specifically targets this generalization: the matrix V is constructed from feature differences of style tokens, ensuring the design optimizes for distinguishing style variations that transfer across content.
>
> ---
>
> We hope this addresses the concerns and would welcome further discussion.

---

### Official Review · Reviewer_EPjt · 2025-11-01

**Soundness:** 3
**Presentation:** 4
**Contribution:** 3
**Rating:** 6
**Confidence:** 4

**Summary:**

This paper looks at the problem of query selection for preference learning. A novel approach rooted in optimal experimental design is presented in this paper. The main contributions are (1) upper bound on MSE as a function of Fisher Information Matrix and converting the query design problem into an optimization problem, (2) operationalizing the optimization problem by reformulating the objective to a tractable form and proposing a novel algorithm to optimize it, (3) experiments using a text-to-image generative model with both a synthetic and human-specified reward model.

**Strengths:**

Query selection is an important problem, so the problem motivation is strong. The idea around using optimal experiment design for preference learning also seems interesting. The theoretical bounds for regularized Bradley-Terry model with a generalized linear reward model in terms of the Fisher Information Matrix could be of more general interest. I also appreciate the effort the authors put in conducting a study with human participants.

**Weaknesses:**

My main concerns are regarding the practicality of the proposed approach. The objective function requires computing the state visitation measures, which is very data hungry especially for high-dimensional state spaces which are usually the norm for generative models like LLMs. Hence the statistical guarantees don't scale well with state space. In addition the objective function requires inverting a matrix, which is computationally very expensive for larger reward models. The experiments are somewhat simplistic with simple reward models, and doesn't provide strong evidence for establishing the practicality of the algorithm. I recommend the authors either address these issues or add a section on the practical limitations of the approach.

**Questions:**

I'm not super familiar with optimal experimental design but a quick search for preference learning with optimal experimental design shows the following papers:
1. Mukherjee, Subhojyoti, et al. "Optimal design for human preference elicitation." Advances in Neural Information Processing Systems 37 (2024): 90132-90159.
2. Schlaginhaufen, Andreas, Reda Ouhamma, and Maryam Kamgarpour. "Efficient Preference-Based Reinforcement Learning: Randomized Exploration Meets Experimental Design." arXiv preprint arXiv:2506.09508 (2025).

Please include these and any other omitted citations if they are relevant to this paper.

---

> ### Author Response · Authors · 2025-12-02
> **Response to Reviewer EPjt**
>
> We thank the reviewer for the thoughtful feedback and for recognizing the importance of query selection for preference learning and the interest of our theoretical bounds based on the Fisher Information Matrix. We address each concern below.
>
> ---
>
> ### Regarding Practicality: State Visitation Measures and Scalability
>
> The reviewer raises an important point about computing state visitation measures in high-dimensional state spaces. We clarify that our formulation does not require enumerating the full state space:
>
> **State visitation measures can be approximated via parametric models.** Just as any probability distribution can be approximated by a parametric family (e.g., via the reparametrization trick), the state-action visitation distribution induced by a policy can be represented implicitly through a parameterized policy network—or even an LLM for text generation. Optimization over the state-action polytope then reduces to optimizing the parameters of this generative model to maximize the linear oracle in the Frank-Wolfe algorithm.
>
> **Our experiments use CLIP embeddings of dimension 768.** The feature dimension d determines the matrix inversion cost, which is O(d³). Modern embedding models (CLIP, SigLIP, etc.) typically use d ≤ 2048, making this inversion highly tractable—a 2048×2048 matrix inverse takes milliseconds on standard hardware. The cost is independent of the raw state space dimensionality (e.g., image pixels or token vocabulary size). We will clarify this computational aspect in the paper.
>
> **The reviewer is correct that the methodology is computationally more expensive than random sampling.** This overhead is justified in settings where human feedback is the bottleneck—each preference query may require significant annotator time and cost, making sample efficiency paramount. Our experiments demonstrate that ED-PBRL achieves equivalent accuracy with substantially fewer queries, which can translate to meaningful cost savings in practice.
>
> We will add a discussion of practical limitations and computational considerations to the revised paper.
>
> ---
>
> ### Regarding Matrix Inversion Cost
>
> To clarify: the matrix being inverted is the d×d Fisher Information Matrix, where d is the embedding/feature dimension, not the state space size. In our experiments, d=768 (CLIP embeddings). Even for the largest embedding models, d rarely exceeds 2048. Therefore:
>
> - Matrix inversion cost: O(d³) ≈ O(2048³) ≈ 8.6 billion operations
> - This takes ~10-100 milliseconds on modern hardware
> - The cost is incurred once during policy optimization (Phase 1), not per-query
>
> We will add explicit complexity statements to clarify this point.
>
> ---
>
> ### Regarding Missing Citations
>
> We thank the reviewer for these excellent references.
>
> **Mukherjee et al. (2024), "Optimal Design for Human Preference Elicitation"**: This work is indeed highly relevant. The key difference is that Mukherjee et al. study preference elicitation over a fixed set of K items (answers to L questions), whereas our algorithm operates on *structured sequential feedback* where queries are trajectories generated by policies in an MDP. This enables learning preferences over compositional spaces (e.g., prompts built token-by-token) rather than pre-defined item sets. While the technical challenges differ—they generalize the Kiefer-Wolfowitz theorem to ranked lists; we combine convex RL with FIM optimization—both works share the core insight that experimental design principles improve preference learning efficiency. We will include this citation and discuss the relationship.
>
> **Schlaginhaufen et al. (2025), "Efficient Preference-Based Reinforcement Learning: Randomized Exploration Meets Experimental Design"**: We note that this paper (arXiv:2506.09508) was posted in June 2025, after the ICLR 2026 submission deadline. Nevertheless, it is closely related and we will cite it as concurrent work. Their use of optimal experimental design for batch query selection complements our approach, though they focus on trajectory-pair comparisons with regret guarantees while we focus on K-way preferences with information-theoretic objectives.
>
> We will include both citations in the revised manuscript.
>
> ---
>
> ### Summary
>
> We appreciate the reviewer's constructive feedback. We will:
> 1. Add a section discussing practical limitations and computational costs
> 2. Clarify that matrix inversion is O(d³) where d is the embedding dimension (typically ≤2048)
> 3. Include the two suggested citations with appropriate discussion
> 4. Note that our new LLM-simulated preference experiments (see general response) provide stronger evidence with 10 target styles and systematic evaluation across regularization and training data sizes
>
> We hope this addresses the reviewer's concerns and would be happy to discuss further.

---

### Author Response · Authors · 2025-12-02
**Revised Human Feedback Experiments with Corrected Regularization**

We thank all reviewers for their valuable feedback. We address a critical issue raised by Reviewer mNCP regarding the modest improvements in human evaluation experiments.

### The Issue: Suboptimal Regularization in Original Experiments

Our original human feedback experiments used regularization values λ ∈ {0.01, 0.1}, which we now recognize were suboptimal for the preference learning setting. The synthetic GT experiments used λ=100, but we mistakenly used much smaller values for human feedback, leading to the modest improvements observed.

### New Experiments: Systematic Evaluation Across λ and Training Data Size

To address this, we conducted comprehensive experiments using GPT-4.1-mini as a simulated preference oracle. The LLM receives identical questionnaire-style queries as human participants—four candidate images and a style description—making this evaluation directly comparable to human studies while enabling systematic evaluation at scale.

**Experimental Setup:**
- **10 target styles:** futuristic, sunny, forest, landscape, ancient, noir, watercolor, cyberpunk, minimalist, medieval
- **Regularization:** λ ∈ {0.1, 1, 10, 100}
- **Training episodes:** {10, 20, 30, 40, 50} with 10 held-out test episodes
- **Cross-validation:** 3 folds per configuration

### Key Results

**Effect of Training Data Size at λ=100** *(see revised PDF, Figure 4a)*

ED-PBRL achieves ~60% accuracy with as low as 10 training episodes, while Random degrades significantly with less data:

| Training Episodes | ED-PBRL | Random | Gap |
|-------------------|---------|--------|-----|
| 10 | 58.6% | 40.1% | **+18.4%** |
| 20 | 59.4% | 45.6% | +13.8% |
| 30 | 60.8% | 47.8% | +13.0% |
| 40 | 60.8% | 49.4% | +11.4% |
| 50 | 60.4% | 51.1% | +9.4% |

**Summary at 50 Training Episodes** *(see revised PDF, Figure 4b)*

At λ=100, ED-PBRL achieves **60.4%** held-out accuracy vs **51.1%** for Random (+9.4pp gap), far exceeding the 25% random-guess baseline.

**Per-Style Breakdown** *(see revised PDF, Figure 13 and Table 2)*

ED-PBRL outperforms Random on 7/10 styles. The advantage is particularly pronounced for styles like Cyberpunk (+28.3pp), Ancient (+22.8pp), and Watercolor (+22.2pp).

### Conclusion

The original modest improvements were due to suboptimal regularization (λ=0.01, 0.1), not a limitation of ED-PBRL. With appropriate λ=100:

1. **ED-PBRL's advantage is substantial:** +9.4% at 50 episodes, +18.4% at 10 episodes
2. **ED-PBRL is particularly valuable in low-data regimes:** Design accuracy stays flat while Random degrades
3. **The gap is consistent across most styles**

We have updated the paper with these corrected experiments. Please see the revised PDF for all new figures.

### Reproducibility

All experiments are fully reproducible. We can provide:
- **Raw LLM feedback responses:** The complete set of GPT-4.1-mini preference answers for all 10 styles × 4 λ values × 2 algorithms
- **Model estimation code:** Scripts to train preference models from feedback data
- **Evaluation pipeline:** Code for cross-validation fold generation, held-out accuracy calculation, and figure generation

The entire pipeline—from raw LLM responses through model estimation to final metrics and figures—can be reproduced upon request with the provided code and data.

---

### Note · Program_Chairs · 2026-01-17
**Submission Desk Rejected by Program Chairs**

The following references in this submission do not refer to real documents and/or have major errors in bibliographic information:

 Erdem Biyik, Dylan J Malayandi, and Dorsa Sadigh. Asking for demonstrations or preferences. In Conference on Robot Learning, pp. 1133-1144. PMLR, 2019.